# Assessing carbon storage capacity and saturation across six central US grasslands using data-model integration

Kevin R. Wilcox[1,2], Scott L. Collins[3], Alan K. Knapp[4], William Pockman[3], Zheng Shi[5], Melinda D. Smith[4], Yiqi Luo[6,7]

[1]Department of Ecosystem Science and Management, University of Wyoming, Laramie, WY 82071, USA
[2]Department of Biology, University of North Carolina Greensboro, Greensboro, NC, 27412, USA
[3]Department of Biology, University of New Mexico, Albuquerque, NM, 87131, USA
[4]Department of Biology & Graduate Degree Program in Ecology, Colorado State University, Fort Collins, CO, 80523, USA
[5]Department of Microbiology and Plant Biology, University of Oklahoma, Norman, OK, 73019
[6]Department of Biological Sciences, Center for Ecosystem Science and Society, Northern Arizona University, Flagstaff, AZ, 86011, USA
[7]School of Integrative Plant Science, Cornell University, Ithaca, NY, 14853, USA

*Correspondence to*: Kevin R. Wilcox (k_wilcox@uncg.edu)

**Abstract.** Future global changes will impact carbon (C) fluxes and pools in most terrestrial ecosystems and the feedback of terrestrial carbon cycling to atmospheric $CO_2$. Determining the vulnerability of ecosystems to future changes in C is thus vital for targeted land management and policy. The C capacity of an ecosystem is a function of its C inputs (*e.g.*, net primary productivity – NPP) and how long C remains in the system before being respired back to the atmosphere. The proportion of C capacity currently stored by an ecosystem (*i.e.*, its C saturation) provides information about the potential for long-term C pools to be altered by environmental and land management regimes. We estimated C capacity, C saturation, NPP, and ecosystem C residence time in six US grasslands spanning temperature and precipitation gradients by integrating high temporal resolution C pool and flux data with a process-based C model. As expected, NPP across grasslands was strongly correlated with mean annual precipitation (MAP), yet C residence time not related to MAP or mean annual temperature (MAT). We link soil temperature, soil moisture, and inherent C turnover rates (potentially due to microbial function and tissue quality) as determinants of $\tau_E$. Overall, we found that intermediates between extremes in moisture and temperature had low C saturation, indicating that C in these grasslands may trend upwards and be buffered against global change impacts. Hot and dry grasslands had greatest C saturation due to both small C inputs through NPP and high C turnover rates during soil moisture conditions favorable for microbial activity. Additionally, leaching of soil C during monsoon events may lead to C loss. C saturation was also high in tallgrass prairie due to frequent fire that reduced inputs of aboveground plant material. Accordingly, we suggest that both hot, dry ecosystems and those frequently disturbed should be subject to careful land management and policy decisions to prevent losses of C stored in these systems.

# 1 Introduction

In the coming decades, most terrestrial ecosystems will experience changes in environmental drivers, including increased air temperatures and atmospheric $CO_2$ concentrations, altered precipitation amounts and patterns, changes in fire frequency, and various anthropogenic impacts (*e.g.*, agriculture)(IPCC, 2022). These changes are likely to have strong impacts on ecosystem functioning, such as C assimilation via plant growth or C losses via respiration (Hungate et al., 1997, Wang et al., 2016, Naylor et al., 2020). These will in turn affect critical ecosystem services, such as C sequestration (Lal, 2004, Wiesmeier et al., 2019). These effects are particularly important in grassland ecosystems due to their global extent (White et al., 2000) and their ability to be sinks for soil C (Conant et al. 2016, Bai and Cotrufo, 2022). Information about grasslands that may experience substantial changes in C storage when subjected to future environmental change is important for targeted land management (Rees et al., 2005) and policy decisions (Daily et al., 2009; Chambers et al., 2016). Experimental studies offer a way to assess how global changes are likely to impact ecosystem processes. Yet, experiments often have difficulty tracking effects on C storage since changes in soil C pools can take decades (Balesdent et al., 1988; Chapin et al., 2011), and most experiments are conducted for relatively short time periods. Process-based models offer another method to assess alterations in soil C under future conditions and have been shown to be useful tools to assess soil C across grasslands (Parton et al., 1993, Bonan et al. 2013). Yet, variation of ecosystem properties and processes controlling C cycling across ecosystems, such as microbial community composition, is not well represented in many current models. Additionally, uncertainty surrounding ecosystem C is currently very large (Todd-Brown et al., 2014, Friend et al., 2014, Luo et al., 2015, Sulman et al., 2018). This highlights the need for better understanding of how C processes vary across ecosystems.

Many estimates of C sequestration rates use two or more time points of C pool measurements to infer annual rates of C accumulation or loss in ecosystems (e.g., Sperow et al. 2016, Smith et al. 2005). While informative, these estimates of C flux rates will not extend indefinitely (Smith 2004), likely due to the non-linear nature of C accumulation or loss through time. Luo et al. (2017) introduced C capacity ($X_C$; Table 1) as the amount of C that would be stored in soil and vegetation in an ecosystem if given enough time to reach equilibrium under current environmental conditions. Comparisons of C capacity and the amount of C currently stored by that system allows for predictions of long-term trends of system C and identification of ecosystems that are vulnerable to C loss under global change. In most terrestrial ecosystems, C capacity is primarily a function of C inputs (NPP) and the amount of time that carbon remains in a system before being respired back to the atmosphere (ecosystem C residence time - $\tau_E$, Luo et al., 2017). There are often mismatches between the amount of C currently present within ecosystems and a system's C capacity because recovery from previous disturbances/environmental conditions can take decades or centuries (e.g., tillage, Smith, 2014). This may underlie observations of grasslands acting as strong C sinks (Soussana et al., 2007). The long-term trajectory (e.g., gains or losses) of C in an ecosystem can be inferred through a comparison of its current C storage at present ($X_P$) with its C capacity (Fig. 1a). It is important to note that the factors influencing C capacity – NPP and C residence time – are constantly changing, and if these changes cause mismatches between present C and C capacity, this will likely alter long-term C trajectories (Fig. 1b). Alternately, C trajectories where

present C is far below capacity may be less vulnerable to global change scenarios if present C remains below future capacity (Fig. 1c). Therefore, we suggest that the proportion of C capacity that is currently present in ecosystems (hereafter termed C saturation – C$_{SAT}$) may be used as an indicator of how vulnerable C pools are to future changes in environmental drivers.

**Table 1. Focal terms, descriptions, and calculation methods used in this study.**

| Symbol | Term | Units | Description | Method of calculation |
|---|---|---|---|---|
| $\tau_E$ | Ecosystem C residence time | year | The average amount of time between fixation of a single C molecule and respiration from the soil. | Integrates residence times of six carbon pools, transfer coefficients among pools, soil moisture, soil temperature, and sensitivity of turnover rates to temperature and moisture (Eqn. 1-5). Here, uncertainty of all the above parameters is integrated into estimates of ecosystem C residence time through bootstrapping methods. |
| NPP | Net primary productivity | g C m$^{-2}$ year$^{-1}$ | The quantity of C produced by plants in one year | Modeled using climate forcing data, benchmarked to empirical observations. |
| X$_C$ | Carbon capacity | g C m$^{-2}$ | The amount of carbon the ecosystem will contain under continuing steady-state conditions | Multiplication of ecosystem C residence time and net primary productivity. We bootstrapped these estimates to incorporate uncertainty in NPP and ecosystem C residence time. |
| X$_P$ | Present carbon | g C m$^{-2}$ | How much carbon is currently present in the system | Sum of C in aboveground plant biomass, belowground plant biomass, and the soil. All estimates were based on empirical measurements but extrapolated to 0-20 cm depths in the soil. |
| C$_{SAT}$ | Carbon saturation | % | The proportion of carbon saturation that is currently present in the system | Carbon capacity divided by the amount of present carbon in the system. |
| F$_T$ | Temperature scalar (associated with $Q_{10}$ model parameter) | -- | Modifies the base C turnover rate dependent on soil temperature | Calculated using soil temperature measurements and the $Q_{10}$ parameter, which is estimated based on empirical data during the data assimilation process. |
| F$_W$ | Moisture scalar (associated with *mscut* model parameter) | -- | Modifies the base C turnover rate dependent on soil moisture | Calculated using soil moisture measurements and the *mscut* parameter, which is estimated based on empirical data during the data assimilation process. |
| $\xi$ | Environmental scalar | -- | Modifies base C turnover rates based on soil | The effect of water (F$_T$) multiplied by the effect of temperature (F$_W$) on C turnover. |

| | | | temperature and soil water content | |
|---|---|---|---|---|

Geographic patterns of C capacity depend on how its components (NPP and C residence time) vary across ecosystems and environmental gradients. There is robust evidence showing patterns of ANPP along gradients of mean annual precipitation (MAP; Sala et al., 1988, 2012, Burke et al. 1997, Huxman et al., 2004, Maurer et al. 2020). Yet, root:shoot ratios may be greater in drier ecosystems (Schenk and Jackson, 2002; Zhou et al. 2009, Mokany et al., 2006; Wilcox et al., 2016, Hu et al., 2022), which may result in shallower relationships between MAP and total NPP. Biomass turnover is associated with C residence time and has been shown to be an important part of biogeochemical responses to changes in environmental conditions (De Kauwe et al., 2014). Patterns of turnover of plant biomass have been linked with numerous drivers in grasslands, including average temperature (Gill and Jackson 2000), precipitation (Yahdjian et al. 2006), tissue quality (Adair et al. 2008), microbial and fungal decomposer communities (Williams and Rice, 2007; García-Palacios et al., 2016), disturbance (Lorenz and Lal, 2018), and often with interactive effects (Bontti et al. 2009). Yet, our understanding is often clouded by abundant contingencies associated with these patterns, effectively limiting our ability to predict which ecosystems will continue to sequester or release C.

Here, we endeavor to generate process-based understanding about how and why C inputs (e.g., NPP) and losses (C turnover) differ among grassland ecosystems, and then use this understanding to identify grasslands where C losses may occur in the future. Integration of data and models (i.e., data-model fusion) is a powerful approach that allows for improved model performance and better estimations of difficult to measure ecological processes and properties (Chen et al. 2010, Fer et al. 2021). Here, we use data-model fusion to (a) assimilate C pool and flux data from six US grassland sites with a process-based ecosystem model (see methods for in depth description of the model) to estimate primary C inputs (NPP), C residence time, and C saturation; and (b) compare present C – the sum of soil and vegetative C – with C capacity to quantify what proportion of C capacity was currently present in each of these ecosystems. With this approach, we address the following questions and predictions:

1. How do NPP, C residence time, and present C vary across gradients of MAP and MAT? We predict that NPP should be primarily related to precipitation since much previous work has shown strong water limitation in grasslands (Sala et al., 1988, Huxman et al., 2004, Maurer et al. 2020), and C residence time will be related to both temperature and precipitation due to strong limitation of these factors on microbial activity.
2. How sensitive is C capacity to turnover rates of different C pools? We predict that changes in turnover rates within slower C pools will have larger effects than faster C pools.

3. Is the amount of C present in any of these systems close to their C capacity? We predict that cooler and drier ecosystems will be further from their C capacity. Ecosystems with low moisture and colder temperature have lower productivity and slower turnover of C pools, both of which can slow the rate that present C approaches C capacity.

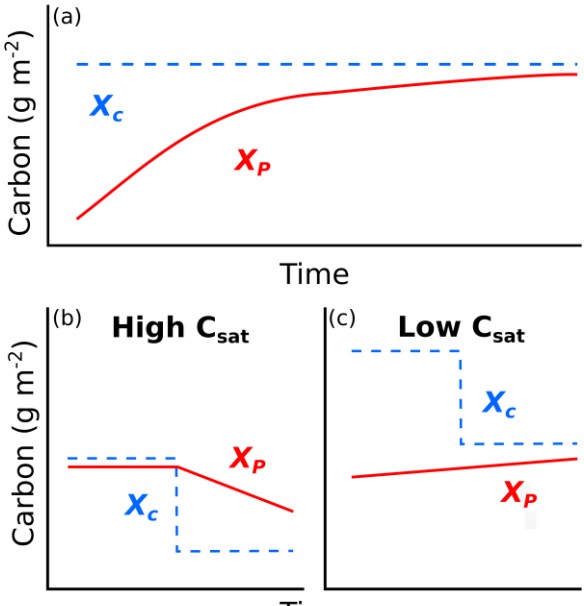

Addressing these questions and predictions will provide an initial perspective on how much these key C attributes vary spatially, as well as identify regions and ecosystems that are vulnerable to C loss, and land areas that should be high priority for future research and management efforts.

**Figure 1. (a) Conceptual figure showing how carbon (C) changes through time to approach C storage capacity as a function of the difference between C storage at present ($X_P$) and C storage capacity ($X_C$). Changes in C capacity (blue dashed lines in b-c) can be caused by alterations in net primary productivity or ecosystem C residence time. (b) Ecosystems that have present C close to capacity are susceptible to C loss if environmental conditions cause reductions in C capacity, while (c) ecosystems having present C far below capacity may be buffered against C losses, at least in the short-term.**

## 2. Methods

### 2.1 Site descriptions

We conducted this study at six US grassland sites spanning climatic gradients of mean annual precipitation (MAP) and mean annual temperature (MAT; Table 2). Data collection sites were set up and maintained as part of the Extreme Drought in Grasslands Experiment (EDGE) and represent the major grassland types within the central United States: desert grassland (SBK), shortgrass prairie (SBL, CPER), northern mixed grass prairie (HPG), southern mixed grass prairie (HAR), and

tallgrass prairie (KNZ). All sites were ungrazed for at least 10 years before the start of data collection, yet the sites did vary in the length of time between the first year of our measurement and when they were last grazed (SBL and SBK: 39 years, CPER: 15 years, HPG: 10 years, HAR: 9 years, KNZ: at least 30 years). All sites except KNZ were not frequently burned, but KNZ was burned annually to reflect common management in this region (Knapp et al. 1998, Freckleton et al. 2004). See Table 2 and Appendix E for more information about these sites.

**Table 2. Site characteristics of each of the six grassland sites in this study.**

| | Site characteristic | SBL | SBK | CPER | HPG | HAR | KNZ |
|---|---|---|---|---|---|---|---|
| Climate[a] | Grassland type | Shortgrass prairie | Desert grassland | Shortgrass prairie | Mixed-grass prairie | Mixed-grass prairie | Tallgrass prairie |
| | Mean annual precipitation (mm) | 246 | 246 | 375 | 400 | 584 | 892 |
| | Mean growing season precipitation (mm)[b] | 163 | 163 | 293 | 303 | 426 | 652 |
| | CV of growing season precipitation[b] | 48.5 | 48.5 | 33.5 | 32.8 | 34.7 | 29.8 |
| | Mean annual temperature (°C) | 13.4 | 13.4 | 9.5 | 7.9 | 12.3 | 13.0 |
| | Mean growing season temperature (°C)[b] | 19.3 | 19.3 | 16.4 | 14.6 | 20.8 | 21.4 |
| Soil | Bulk density (g cm$^{-3}$)[c] | 1.68 | 1.68 | 1.26 | 1.18 | 1.16 | 1.03 |
| | Field capacity (% soil moisture)[d] | 27 | 30 | 17 | 29 | 35 | 38 |
| | Wilting point (% Soil moisture)[d] | 7 | 5 | 5 | 10 | 16 | 15 |
| Vegetation[e] | C$_3$ graminoid (%) | 0 | 0 | 17.5 | 53.7 | 9.0 | 11.9 |
| | C$_4$ graminoid (%) | 48.7 | 52.0 | 54.0 | 27.0 | 68.9 | 77.0 |
| | CAM (%) | 22.8 | 0 | 6.6 | 0 | 0.4 | 0 |
| | Forb (%) | 24.2 | 44.86 | 19.2 | 12.5 | 19.6 | 8.4 |
| | Woody (%) | 3.4 | 2.3 | 1.9 | 5.9 | 1.2 | 2.5 |
| | Perennial (%) | 82.4 | 77.5 | 82.5 | 95.1 | 96.6 | 99.5 |
| | Annual (%) | 16.8 | 21.5 | 16.6 | 3.9 | 2.4 | 0.4 |

[a]Climate characteristics are from 1982-2012 weather data, obtained from Knapp et al. 2015
[b]Growing season was defined as April-September for CPER, HPG, HAR, and KNZ, and as April-October for SBL and SBK
[c]Bulk density data obtained from measurements taken at each site in 2015
[d]Estimated using hourly soil moisture data from 2012-2013 in SBL and SBK and from 2013-2015 in CPER, HPG, HAR, and KNZ
[e]Estimated from plant species composition measurements taken during the 2012-2013 growing seasons at SBL and SBK and during the 2014 growing season at CPER, HPG, HAR, and KNZ

**2.2 Sampling design**

For this study, we used measurements of aboveground net primary productivity (ANPP), belowground net primary productivity (BNPP), root standing crop biomass, vegetative litter biomass, soil C, volumetric soil moisture, soil temperature, soil $CO_2$ efflux, plant species abundance, soil bulk density, and hourly meteorological data. Most of these data were collected from control plots within experimental infrastructure, which is a randomized block design having 10 blocks each containing three treatments: one control and two drought treatments; for the purposes of this study, we only use control

data from the ten 6 $m^2$ control plots at each site. See Appendix C for additional details about sampling regimes.

**2.3 Estimating GPP and NPP**

To generate gross primary productivity (GPP) and net primary productivity (NPP) estimates, we operated the grassland version of the Terrestrial Ecosystem Model (TECO; Weng and Luo, 2008; Shi et al., 2015), which has been shown to produce C flux estimates that match observations well in US grassland ecosystems (Shi et al., 2014). TECO is a process-

based ecosystem model that has four major sub-models to simulate canopy photosynthesis, plant growth (allocation and phenology), soil water dynamics, and soil carbon turnover based on weather data and site level soil characteristics (Figure A1). To run the model, we used hourly air temperature, relative humidity, vapor pressure deficit, precipitation, and incident photosynthetically active radiation data from nearby weather stations (see Appendix C for additional details about collecting and cleaning meteorological data). GPP and NPP were generated for the main analyses in this paper using TECO for 2012-

2014 at SBL and SBK, and for 2013-2015 at the other four sites. Daily GPP estimates were subsequently used to drive the C sub-model (section 2.4), and annual NPP estimates were used to calculate C capacity (section 2.5). The mismatch in time frame among sites was due to data availability.

Formal validation of the vegetation components of the model was conducted at each of the six sites. This was done by calibrating the model for each site based on measured above and belowground plant growth, soil texture, site-level field

capacity and wilting point. Then, model spin-up of 500 years (all pools stabilized at each site between 200 and 400 years) was conducted and output from 2014-2017 was compared with observations at each site. Overall, cross-site mean primary production estimates from the model matched empirical observations very well (aboveground biomass $R^2$=0.99, RMSE 18.0; belowground NPP $R^2$=0.94, RMSE=23.5). Interannual variability in production from the model was less well correlated with empirical observations, although model predictions most often fell within one standard deviation of empirical observations.

(See Appendix D for additional model validation discussion, figures, and tables).

**2.4 Optimizing C sub-model parameters**

Within the C turnover sub-model in TECO (Fig. A1), parameters for C turnover rates, C transfer rates, and environmental scalars (Table B1) were estimated for each site using data assimilation techniques (Xu et al., 2006; Shi et al., 2015).

Compared with benchmarking, this is a more powerful approach for improving model parameterization, but it often requires sufficient temporal resolution and richness of data describing multiple components of modeled variables to be successful. We used estimated daily GPP, soil moisture, and soil temperature to operate the C sub-model within the data assimilation procedure to optimize the following sets of parameters: (1) six C turnover parameters associated with leaf, fine root, litter, fast SOM, slow SOM, and passive SOM carbon pools, (2) seven C transfer coefficients controlling the proportion of C turnover transferred to other C pools, and (3) two environmental scalars that control C turnover rates based on soil moisture and soil temperature (Table B1). We used a Markov Chain Monte-Carlo method with Metropolis-Hastings algorithm to optimize these parameters. Starting parameter values were obtained from previous studies (Xu et al., 2006, Shi et al., 2015, Zhou et al., 2012) and were allowed to vary uniformly between biologically reasonable bounds (Table B1). Within each iteration, the current set of parameters was tested against a new set of parameters, generated based on the current set of parameters using a step size of 15 with the Metropolis-Hastings Algorithm. Both the current and new set of parameters were used to run the C sub-model with daily GPP estimates (section 2.3), daily measured soil moisture, and daily measured soil temperature from each site. Model output from each of these two runs was then compared with the observations of aboveground vegetation biomass (annually), root standing crop (annually), plant litter (annually), soil C (single measurement), and surface $CO_2$ efflux (daily). Model performance using the new set of parameters assessed against Metropolis criterion to determine whether the new set of parameters should be kept or discarded. This was done for 360,000 iterations for each of 4 chains within each site to ensure convergence of parameter estimates. Gelman-Rubin (GR) values were mostly < 1.1, with the exception of a few parameters having 1.2 or 1.3 GR values at HPG and HAR (Table B2). All parameters where GR values were high did not converge and drifted slowly over iterations. This resulted in estimates of these parameters close to the midpoint of the parameter bounds, and large uncertainty. To account for this, uncertainty in parameter estimates was incorporated into C residence time estimates via bootstrapping methods (see below). Cross-correlations were calculated for all parameters at each site (Table B3). Maximum likelihood estimates (MLE) and uncertainty (95% confidence intervals) were calculated for each parameter at each site by assessing normal, log-normal, or Weibull distributions depending on the magnitude and direction of skew (Fig. A2-A4, Table B2).

**2.5 Estimating C residence time, C capacity and C saturation**

We calculated ecosystem C residence time ($\tau_E$) following Luo et al. (2017):

$$\tau_E = (A\xi(t)K)^{-1}B \tag{1}$$

, where $\xi(t)$ represents the environmental scalar determined by soil moisture and soil temperature at time step *t, A* is a matrix of C transfer coefficients, *K* is a 6x6 diagonal matrix representing rates of C loss per day from each of the six C pools, and B is a 6x1 matrix representing the allocation fractions of GPP to each of the six C pools:

$$\xi(t) = F_T(t)F_W(t); \tag{2}$$

$$A = \begin{pmatrix} -1 & 0 & 0 & 0 & 0 & 0 \\ 0 & -1 & 0 & 0 & 0 & 0 \\ 1 & 1 & -1 & 0 & 0 & 0 \\ 0 & 0 & f_{4\leftarrow3} & -1 & f_{4\leftarrow5} & f_{4\leftarrow6} \\ 0 & 0 & f_{5\leftarrow3} & f_{5\leftarrow4} & -1 & 0 \\ 0 & 0 & 0 & f_{6\leftarrow4} & f_{6\leftarrow5} & -1 \end{pmatrix} : \tag{3}$$

$$K = \begin{pmatrix} c_1 & 0 & 0 & 0 & 0 & 0 \\ 0 & c_2 & 0 & 0 & 0 & 0 \\ 0 & 0 & c_3 & 0 & 0 & 0 \\ 0 & 0 & 0 & c_4 & 0 & 0 \\ 0 & 0 & 0 & 0 & c_5 & 0 \\ 0 & 0 & 0 & 0 & 0 & c_6 \end{pmatrix} \tag{4}$$

$$B = (X_A, X_B, 0, 0, 0, 0) \tag{5}$$

In Eqn. 2, $F_T$ is the effect of soil temperature on microbial decomposition rates at time $t$: $Ft(t) = 0.58Q_{10}^{(T(T)-10)/10}$, where $Q_{10}$ is a constant parameter and $T$ is soil temperature. $F_W$ is the potential effect of soil water content on microbial decomposition

rates at time $t$: $F_W(t) = 1 - 5(mscut-W(t))$, where *mscut* is a constant parameter representing the soil water content ($W$) below which microbial decomposition becomes limited. If $W$ is greater than *mscut*, $F_W = 1$. The impact of $F_T$ and $Fw$ scalars on ξ are dependent on one another (*i.e.*, $F_W$ will limit ξ in dry conditions even if soil temperatures lead to a large $F_T$). In Eqn. 3, $f_{i\leftarrow j}$ represents the fractions of C turnover entering pool $i$ pool from pool $j$. In Eqn. 4, $c_{1-6}$ represents the amount of carbon lost from pools 1-6 per day, where pool 1=aboveground plant biomass, 2=belowground plant biomass, 3=fine litter biomass,

4=active (fast) soil organic matter (SOM), 5=slow SOM, 6=passive SOM. In Eqn. 5, $X_A$ and $X_B$ are the fractions of GPP allocated to aboveground and belowground vegetative pools, respectively. For each site, $X_A$ and $X_B$ were estimated from observed ANPP:BNPP ratios, and data assimilation was used to estimate $c_{1-6}$, $f_{i\leftarrow j}$, $Q_{10}$, and *mscut* parameters. To generate uncertainty surrounding $\tau_E$, we bootstrapped 1000 parameter sets from the Markov chain Monte Carlo (MCMC) and obtained the MLE and 95% confidence intervals from the resulting distribution of $\tau_E$ estimates.

C capacity ($X_C$) was calculated following Luo et al. (2017) as:

$$X_C = NPP \cdot \tau_E \tag{6}$$

, where NPP is net primary productivity of a site, obtained via TECO simulations, and $\tau_E$ is the MLE of the distribution of bootstrapped $\tau_E$ values. C capacity estimates were obtained by combining the boostrapped iteration of C residence estimates with 1000 randomly sampled values of NPP using the mean and standard deviation of NPP across years. This allowed us to

205 propagate the uncertainty present in both NPP and C residence time to C capacity estimates. The 1000 boostrapped iterations were then used below in the calculation of C saturation.

At KNZ, NPP in Eqn. 6 consisted only of the belowground component because annual fire removes all aboveground plant material each spring. We recognize the limitation of using three years of NPP data to estimate $X_C$, yet we believe it is important that NPP and $\tau_E$ estimates are derived from the same time periods, and the data necessary to estimate $\tau_E$ were only available for three years. Weather within the three focal years was comparable to long-term averages at most of the sites (Fig. A5), although precipitation was greater than the long-term average at HPG, lower than the long-term average at HAR, and air temperatures were warmer at SBK and SBL in 2012-2014. Standard deviation of C residence time ($C_{sd}$) was calculated as the standard deviation of the 1000 bootstrap iterations.

The level of C saturation ($C_{SAT}$) represents the percentage of C capacity that is represented by present C, calculated as

$$C_{SAT} = \frac{C_S + C_A + C_B}{X_C} \tag{7}$$

, where $C_S$ is the mass of C in the soil standardized by area, $C_A$ is the observed aboveground biomass * 0.45, and $C_B$ is the observed root biomass * 0.45, also standardized by area. Combined, $C_S$, $C_A$, and $C_B$ make up present C from 0-10 cm in the soil. Soil C measurements from 0-10 cm in the soil were then extrapolated to 0-20 cm to match up with the depth of BNPP observed and used to calibrate the model. This was done by extracting soil C data along a depth profile (0 to >1 m depth) from the international soil carbon network (ISCN; Nave et al 2017) in nearby areas having similar cover types and land management regimes (Table B4). These depth profiles were used to calculate the proportion of soil C across depths using a beta distribution described by Jobággy and Jackson (2000)(Fig. A7). Then, each soil C measurements from 0-10 cm was extrapolated along this curve to estimate the amount of soil C from 0-20 cm (Fig. A7). The mean and standard deviation among replicates within a site were used to generate 1000 random draws from a normal distribution. We then combined these random draws with the bootstrap iterations from Eqn. 6 to propagate the uncertainty of C capacity ($X_C$) into the estimate of C saturation. This means that all levels of uncertainty, from individual parameter estimates (Fig. A2-A4) all the way through present C are incorporated into our estimates of C saturation.

We conducted variance partitioning to determine the amount of cross-site variance in C capacity that was driven by variation in NPP versus C residence time. Since only BNPP was incorporated into the $X_C$ calculation for KNZ, we performed this analysis both with and without KNZ (Fig. A6).

**2.6 Sensitivity analyses**

For each parameter used to calculate C residence time, we varied the parameter while keeping all other parameters constant at their MLE and recorded the resulting C residence time. We did this for 20 intervals ranging from the minimum to maximum parameter values shown in Table B1. We also wished to determine the impact of each parameter value at each site as estimated via data assimilation. To this end, we shifted each parameter from its default value (Table B1) to the MLE value

obtained from data assimilation (Table B2) – holding all other parameters at their default values – and observed the resulting effect on C residence time (Fig. 3g-l).

## 2.7 Statistical analyses

For regression analysis comparing NPP, C residence time, and present C across gradients of MAP, and MAT, all variables
were centered by their mean and scaled by their standard deviation, allowing for comparable slope values. Additional site-level characteristics (Bulk density, grass:forb, $C_3$:$C_4$, Annual species abundance) were combined with climate data using partial regression and adjusted $R^2$ values were assessed to test whether climate-NPP or climate-C residence time relationships were being driven by other site-characteristics (vegan package, Oksanen et al. 2016). Bayesian data assimilation and bootstrapping analyses were run using custom scripts; linear regression models were run with the lm()
function. All analyses were conducted in R (R core team, 2022).

## 3 Results

### 3.1 Net primary productivity (NPP) and present C ($X_P$)

Estimates of NPP varied across sites from 45.3 g C m$^{-2}$ yr$^{-1}$ at SBL to 400.2 g C m$^{-2}$ yr$^{-1}$ at KNZ (Table 3). The standardized full NPP model (NPP ~ MAP + MAT) was significant ($F_{2,3}$=63.8, P<0.01) and explained 96% of the cross-site variation in
NPP estimates (Adj. $R^2$ = 0.96). Within the model, MAP was strongly correlated with NPP across sites ($F_{1,3}$=123.7, P<0.01), while the relationship with MAT was not significant ($F_{1,3}$=0.58, P=0.50)(Fig. 2). The non-standardized relationship between MAP and NPP was of the form $NPP = 0.53*MAP - 51.4$. We looked for collinearity of MAP with soil bulk density, grass:forb, $C_3$:$C_4$, and annual species abundance using partial regression analysis. We found that MAP was still a significant and strong predictor of NPP when these other variables were accounted for (Table B5). Similarly, we found a weak positive
relationship between present C and MAP ($F_{1,3}$=6.59, P=0.08, Adj. $R^2$=0.54), and no relationship between MAT and present C (Fig. 2). The non-standardized relationship between MAP and present C was of the form $X_P = 767.4 + 4.0*MAP$.

**Table 3. Estimates (μ and q$_{50}$) and uncertainty (σ and other quantiles) of NPP and ecosystem carbon residence time at all six sites.**

| Site | C residence time ($\tau_E$)(years) | | | | | | | NPP (g C m$^{-2}$) | |
|------|------|------|------|------|------|------|------|------|------|
| | q$_{2.5}$ | q$_5$ | q$_{25}$ | q$_{50}$ | q$_{75}$ | q$_{95}$ | q$_{97.5}$ | μ | σ |
| SBL | 5.2 | 7.1 | 18.3 | 35.3 | 68 | 175.1 | 238.1 | 45.3 | 15.3 |
| SBK | 11.4 | 13.2 | 20.4 | 27.8 | 37.7 | 58.5 | 67.5 | 67.7 | 27.6 |
| CPER | 19.1 | 22.5 | 36.8 | 51.8 | 72.9 | 119.2 | 139.9 | 143.6 | 25.0 |
| HPG | 21.2 | 25.2 | 42.5 | 61.3 | 88.2 | 149.2 | 176.9 | 157.6 | 65.2 |
| HAR | 17.3 | 21.1 | 39 | 59.9 | 91.9 | 170.2 | 207.9 | 173.3 | 31.8 |

| KNZ | 9.9 | 11.9 | 20.7 | 30.4 | 44.7 | 77.8 | 93.2 | 400.2 | 141.0 |

## 3.2 Ecosystem carbon residence time ($\tau_E$)

Estimates of C residence time were obtained by calculating the 50[th] percentile of the lognormal distribution of bootstrapped C residence time values. These estimates ranged across sites from 27.8 years at SBK to 61.3 years at HAR (Table 3). The standardized full C residence time model ($\tau_E \sim$ MAP + MAT) was not significant ($F_{2,3}$=2.05, P=0.27). Within the full model,

neither MAP ($F_{1,3}$=0.10, P=0.77) nor MAT ($F_{1,3}$=4.10, P=0.14) were correlated with C residence time across sites (Fig. 2b).

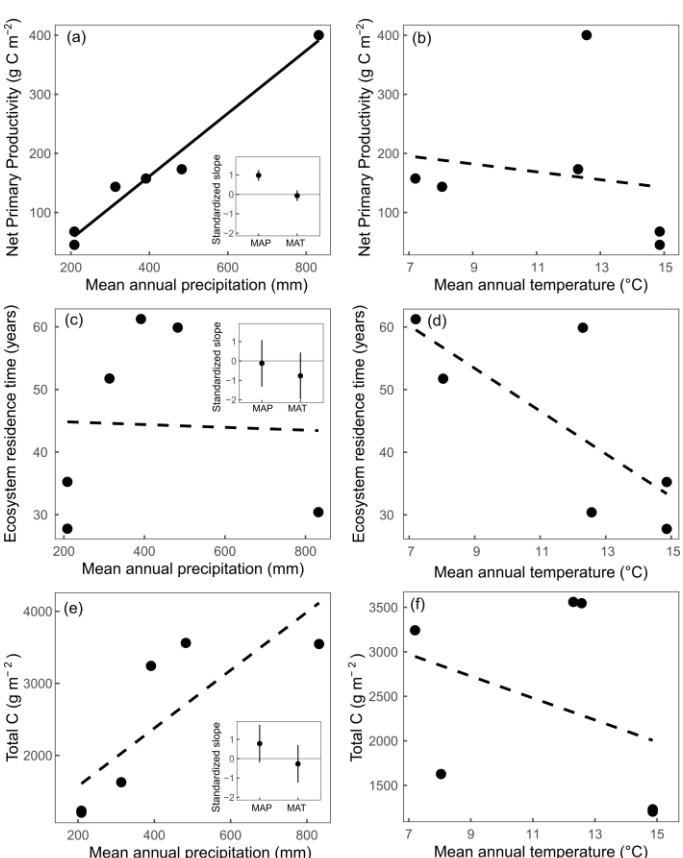

**Figure 2. relationships between mean annual precipitation (MAP) and mean annual temperature (MAT) at a site with net primary productivity (a,b), ecosystem C residence time (c,d), and total C in vegetation and the top 20 cm of soil (e,f).**
**Insets represent the standardized slopes with MAP and MAT with error bars representing 95% confidence intervals, and solid regression lines represent slopes significant at α=0.05. Linear relationships were compared with log-linear relationships in all cases and linear relationships represented the best fit in all scenarios.**

### 3.3 Soil moisture and temperature effects on C residence time

All sites exhibited a cyclical pattern of soil moisture and temperature effects on C turnover rates (denoted $\xi$), with higher $\xi$
during the growing season due to warmer temperatures (Fig. 3 black lines). $\xi$ during the growing season was $> 1$ for all sites
except CPER, meaning that C turnover rates were increased in the data driven model, rather than limited by soil conditions.
At SBL and SBK (Fig. 3a,b), temperature constraints ($F_T$; Fig. 3 dashed orange lines) on $\xi$ were $> 1$ for much of the growing
season, yet $\xi$ was limited by soil moisture constraints ($F_W$; Fig. 3 dotted blue lines) outside of the monsoon season. Only
when monsoon rains removed soil moisture limitations did $\xi$ generally persist above one. At CPER (Fig. 3c), $\xi$ was not
limited by $F_W$. Yet, $F_T$ was $< 1$ throughout the year due to the low $Q_{10}$ value estimated for CPER (Table B2). $F_T$ was much
greater than one during the growing season at both the mixed grass and tallgrass prairie sites (Fig. d-f), but $F_W$ limited $\xi$ at
both mixed grass prairies. $\xi$ was high at KNZ due to a lack of $F_W$ effect at the site, which was a result of both high soil
moisture content throughout the growing season, and a relatively low estimated *mscut* parameter value (Table B2).

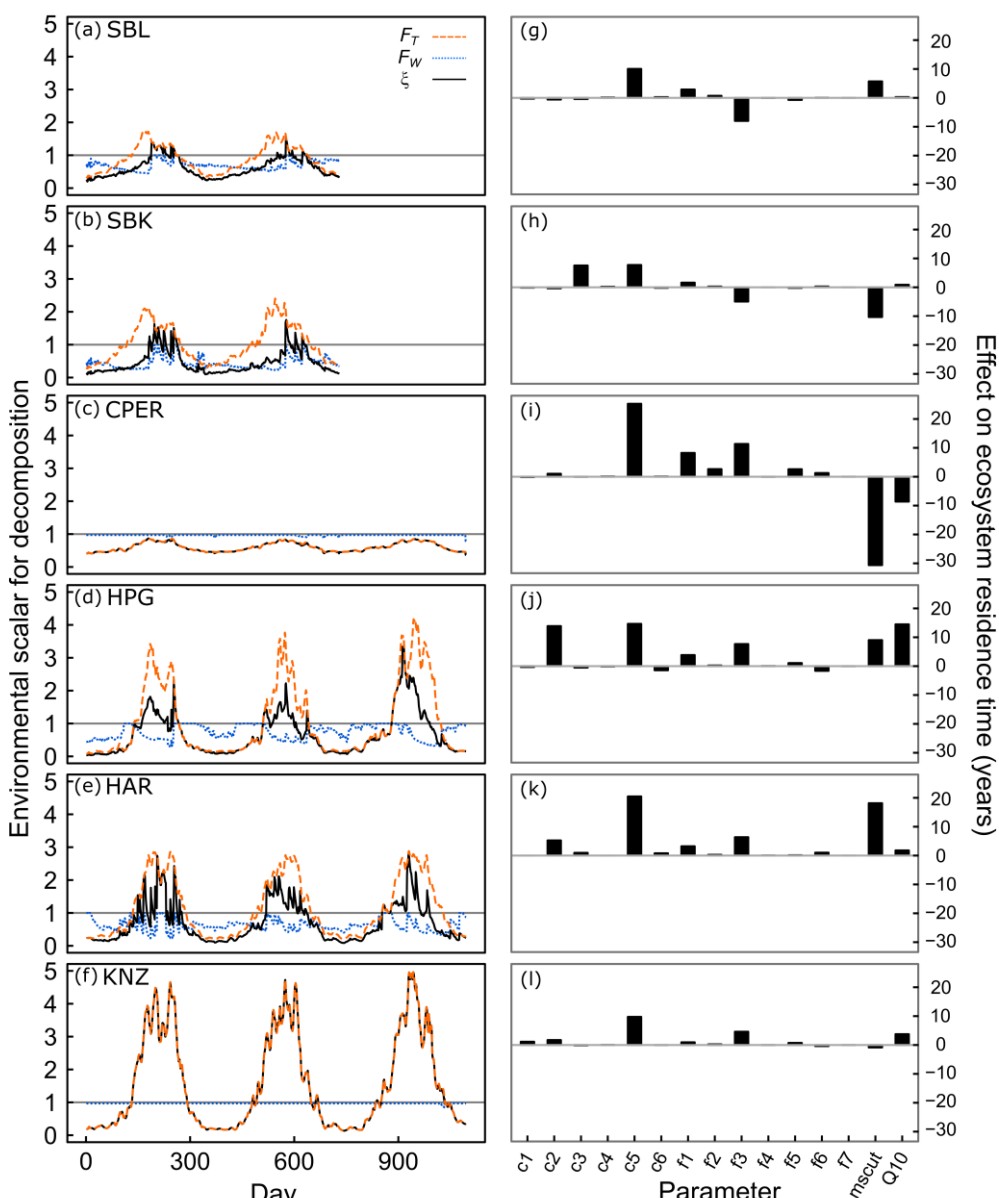

Figure 3. Environmental scalars for decomposition rates at six grassland sites (a-f) and impact of individual parameter estimates on ecosystem carbon residence time (g-l). In a-f, values less than 1 represent soil conditions limiting decomposition while values greater than 1 represent acceleration of decomposition due to soil conditions. Dashed orange lines represent the temperature scaling effect ($F_T$) – based on the site-estimated value of the $Q_{10}$ parameter and daily soil temperature data. Dotted blue lines represent the moisture scaling effect ($F_W$), which is based on the site-estimated value of the *mscut* parameter and daily soil moisture data. Black solid lines represent the product of the temperature and moisture scalars ($\xi$), which is the overall environmental scalar that controls decomposition rates in the model. In g-l, parameters were shifted one at a time from their mean parameter space (baseline parameter) to the parameter estimates obtained from data assimilation, and the resulting effect on ecosystem carbon residence time is

shown. This represents the inherent effect of each model parameter on $\tau_E$ independent from soil moisture or temperature. Panels correspond to different sites: a,g=Sevilleta National Wildlife Refuge blue grama grassland; b,h=Sevilleta National Wildlife Refuge black grama grassland; c,i=Central Plains Experimental Range; d,j=High Plains Grasslands Research Station; e,k=Hays Agricultural Research Center; f,l=Konza Prairie Biological Station. Transfer parameters ($f_{x \leftarrow y}$) dictate the proportion of C turnover in pool $y$ transferring to pool $x$: f1=f4$\leftarrow$3; f2=f5$\leftarrow$3; f3=f5$\leftarrow$4; f4=f6$\leftarrow$4; f5=f4$\leftarrow$5; f6=f6$\leftarrow$5; f7=f4$\leftarrow$6.

**3.4 Sensitivity of C residence time to model parameters**

We performed two sensitivity analyses to (1) identify variables in the model with potential to contribute the most to $\tau_E$ (Fig. 4, Fig. A10), and (2) quantify the realized effect of model parameter estimates, obtained through the data assimilation process, on C residence time at each site (Fig. 3). These sensitivity analyses simulated C residence time under a range of parameter values while incorporating daily soil temperature and moisture measurements from each site. We found that C turnover rates of the slow and passive SOM pools ($c_5$ and $c_6$ model parameters) had the potential to have the greatest impacts on C residence time, highlighting the importance of C sequestration in these pools for maintaining C stocks (Fig. 4, Fig. A10). C residence time was also sensitive to the *mscut* parameter, with the effect increasing exponentially until reaching ca. field capacity at each site. In the drier sites (SBL, SBK, and CPER), the effect of the *mscut* parameter (*i.e.*, the soil moisture percentage at which C turnover begins to be limited) started to increase rapidly around parameter values of 10-15. The steep increase began at greater greater parameter values (20-25) at the more mesic sites (HPG, HAR, KNZ). The $Q_{10}$ parameter describes the sensitivity of C turnover to soil temperature, with greater values indicating faster turnover rates at higher temperatures. At the warmer sites (SBL, SBK, KNZ), C residence time was less sensitive to $Q_{10}$, unless $Q_{10}$ was very low (ca. 1). Interestingly, higher $Q_{10}$ values had the potential to increase C residence time at the cooler sites (CPER, HPG). Although $Q_{10}$ is generally negatively related to C residence time, the form of the $Q_{10}$ relationship is such that, although lower $Q_{10}$ values result in slower turnover rates at high soil temperatures, they also result in higher turnover rates under cooler temperatures due to their shallower slope. As such, this can lead to a positive relationship between $Q_{10}$ values and C residence time at cooler sites.

When we applied the estimated parameters to these sensitivity curves to estimate the actual effect of individual parameters on the C residence time estimates (Fig. 3 right panels), we found that turnover rates of the slow SOM pool increased C residence time greatly compared with starting parameters across all sites (+8 to +25 years). Root turnover had a substantial effect on C residence time at HPG (+14 years) and HAR (+5 years). The transfer proportion from fast SOM to slow SOM had a positive effect on C residence time at CPER, HPG, HAR, and KNZ (+5 to +11 years), and a negative effect on C residence time at SBL and SBK (-5 to -8 years). *mscut* had opposite effects on C residence time for SBL (+6 years) than SBK (-10 years), while $Q_{10}$ had minimal effects at both sites. *mscut* and $Q_{10}$ had strong negative effects on C residence time at CPER (-31 years and -9 years, respectively), likely due to particularly low *mscut* and $Q_{10}$ estimated for that site (Fig. A4) and strong potential for impact of these parameters at CPER (Fig. 3). At the other cool site, HPG, both *mscut* and $Q_{10}$

increased C residence time (+9 and +14 years, respectively). At HAR, a high *mscut* estimate increased C residence time (+18 years), suggesting that C turnover at this site may be particularly sensitive to soil moisture. $Q_{10}$ estimated at HAR had minimal impact. *mscut* and $Q_{10}$ estimates at KNZ had small impacts on C residence time (-1 and +4 years, respectively, Fig. 3).

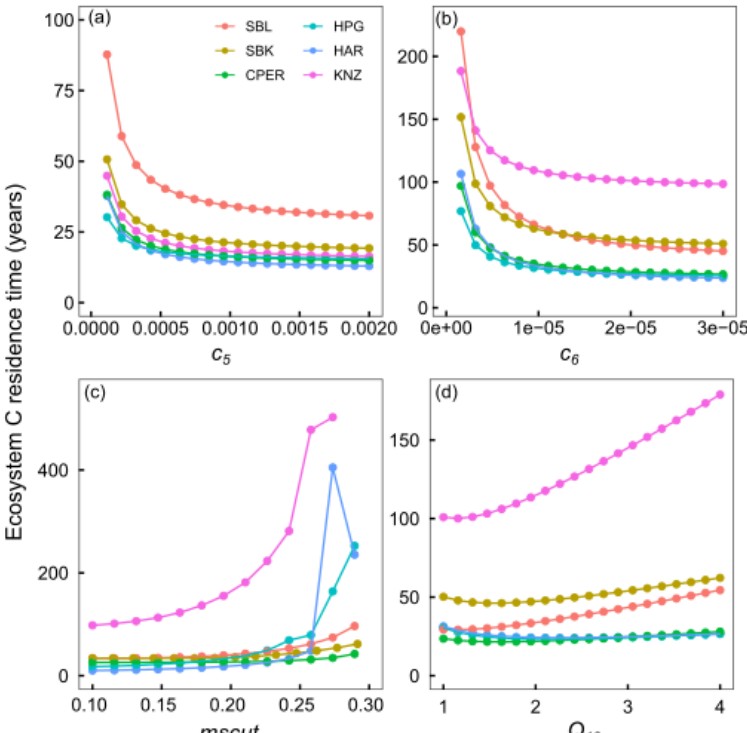

**Figure 4. Results from sensitivity analysis where ecosystem C residence time was calculated when altering one parameter value at a time. Parameters shown here are the turnover rate of the slow C pool (panel a, $c_5$), turnover rate of passive C pool (panel b, $c_6$), a parameter controlling sensitivity of C turnover to soil moisture(panel c, *mscut*), and a parameter controlling sensitivity of C turnover to temperature (panel d, $Q_{10}$). Colors correspond to different sites: SBL= Blue grama dominated site at the Sevilleta National Wildlife Refuge; SBK= Black grama dominated site at the Sevilleta National Wildlife Refuge; CPER=central plains experimental range; HPG=High Plains Grassland Research Station; HAR=Hays Agricultural Research Station; KNZ=Konza Prairie Biological Station. Ecosystem carbon residence time was often very high at very extreme parameter values so the y axis was set for clarity.**

### 3.5 Carbon capacity and carbon saturation

Finally, we used NPP and C residence time estimates to calculate C capacity. Cross-site variation of NPP and C residence time were both important for determining C capacity across the six grassland sites (Fig. A6). Median C capacity varied from

as little as 1485 g m$^{-2}$ in SBL to as much as 10203 g m$^{-2}$ in HAR (Fig. 5). We estimated C saturation as the percentage of C
capacity made up by present C. In the two hot and dry sites (SBL and SBK), we found that C capacity was relatively small
and less than present C (Fig. 5), resulting in greater C saturation values (50$^{th}$ percentiles of C saturation lognormal
distribution: SBL 148%, SBK 130%, Fig. 5 inset). The cooler and/or wetter sites all had greater C capacity values and
present C below capacity (Fig. 5), resulting in smaller C saturation values (50$^{th}$ percentiles: CPER 36%, HPG 56%, HAR
58%, Fig. 5 inset). The exception to this was KNZ, the most mesic but frequently burned site with a C saturation value of
137% (50$^{th}$ percentile, Fig. 5 inset).

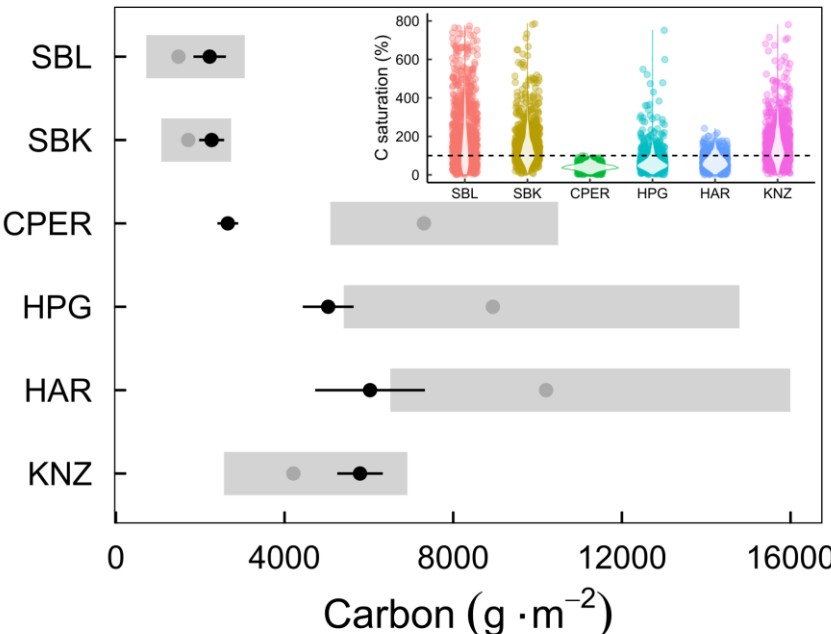

**Figure 5. Ecosystem C capacity ($X_C$; grey points) and present C ($X_P$; black points) in six grassland ecosystems. Present C was calculated using estimates of soil C from 0-20 cm + aboveground and belowground vegetative C. Black error bars represent one standard deviation around the mean, and grey rectangles represent the 25$^{th}$ and 75$^{th}$ percentiles of the lognormal distribution of C capacity, surrounding the 50$^{th}$ percentile (grey points). Inset: Violin plots and bootstrap estimates of C saturation, calculated as present C divided by the product of NPP and ecosystem C residence time. The dashed line shows where present C is equal to C potential.**

## 4. Discussion

Our findings provide insights for the three questions we posed at the start of this study: (1) How do NPP, C residence time, and present C vary across gradients of MAP and MAT? (2) Is present C in any of these systems close to C capacity? and (3)

How does the level of C saturation vary across these grasslands? Related to our first question, we found general support for the prediction that NPP and present C exhibited positive relationships with MAP, yet found no relationship between ecosystem C residence time and climate or other site-level characteristics. Instead, the cross-site pattern of C residence time was driven by differences in local edaphic environments (soil moisture and soil temperature) as well as inherent differences in turnover rates, which may be indicative of biological or physical differences across sites (Baisden et al., 2013, Mathieu et al., 2015, Doetterl et al., 2015, Zhao et al., 2021). Related to our second and third questions, we found that three of these grasslands had particularly high C saturation values, indicating vulnerability to C change in the future and a limited ability of these systems to be long-term C sinks. Two of these three grasslands were in hot, dry climates where C turnover rates were high and C inputs through NPP low. The third grassland (KNZ) was the most mesic and had the highest levels of productivity, yet annual burning at KNZ increased C saturation substantially. Below, we discuss these findings in more detail.

## 4.1 Relationships of NPP, C residence time, and present C with climate

Abundant research exists showing spatial relationships between ANPP and climate. Sala et al. (1988) was able to explain 90% of the cross-site variation in averaged ANPP with mean annual precipitation across the Great Plains. Yet, total NPP (ANPP + BNPP) is a better determinant of C processes due to large contributions of root C to soil pools (Sulzman et al., 2005; Guzman and Al-Kaisi, 2010; Leppälammi-Kujansuu et al., 2014). A potential reason why BNPP and total NPP relationships with climate may be less clear than ANPP relationships is that, in wetter ecosystems, plants tend to allocate less carbohydrates to roots and more to aboveground material (Schenk and Jackson, 2002, Mokany et al., 2006, Zhou et al. 2009, Wilcox et al., 2016, Hu et al., 2022). This pattern results in a weaker relationship between MAP and NPP than predicted by ANPP-MAP relationships since BNPP is proportionally greater in drier ecosystems. Indeed, we found some evidence for this from our model simulations – the slope of the BNPP-MAP regression (0.24 +/- 0.03, slope estimate +/- standard error) was shallower than the slope of the ANPP-MAP regression (0.29 +/- 0.02, Fig. A8). Also, differences in functional composition of vegetation may drive site differences in root:shoot (e.g., annual versus perennial species). Despite the additional uncertainty associated with total NPP, we found that MAP was a strong predictor of total NPP across the six grassland sites (Fig. 2a).

We predicted that C residence time should be greater in (1) cooler systems due to lower soil temperatures and shorter growing seasons and (2) drier systems due to moisture limitations on microbial activity. Previous studies examining patterns of C residence time have found relationships of varying strengths with climate or latitude (Bird et al., 1996, Chen et al., 2013, Carvalhais et al., 2014, Moore et al., 2018), biome type (Zhou and Luo, 2008), soil properties (Telles et al., 2003), vegetation tissue quality (Adair et al. 2008, Bontti et al. 2009), and land use change (Sperow et al., 2016, Wu et al., 2020). Yet, there is still much uncertainty associated with trends in C residence time (Friend et al., 2014). We did not find relationships between any of the course site-level characteristics fo0und that neither MAP nor MAT were good predictors of

C residence time across the six grasslands we examined (Fig. 2b). Instead, it is likely that more nuanced characteristics of sites. SBL and SBK both had particularly short C residence times, likely due to strong C limitation of microbes at these sites and high abundances of fungal decomposers that efficiently break down recalcitrant C (Collins et al., 2008, Sinsabaugh et al., 2008). Additionally, intense wet-dry cycles (Fierer and Schimel, 2002), soil burial (Brandt et al., 2010), and photo-degradation (Austin and Vivanco, 2006, Parton et al., 2007) have all been shown to be important accelerators of

decomposition rates in arid systems and may be contributing to the low C residence times in these grasslands. As soil C is a function of both NPP and C residence time, it makes sense that the stronger relationship of the two is the one that is best related to present C. We found that the best variable related to present C was MAP (Fig. 2), so soil C may be more sensitive to changes in precipitation versus temperature in U.S. grassland systems. This corresponds with observational studies (Saiz et al., 2012) as well as meta-analysis findings of stronger moisture than temperature effects on net ecosystem exchange (Wu

et al., 2011).

### 4.2 Effects of soil environment versus inherent site differences on C turnover rates

    C residence time is directly related to various C turnover rates within an ecosystem (Luo et al., 2017). These turnover rates can be driven by favorability of soil environments for microbial activity (Bird et al., 1996, Carvalhais et al., 2014, Stielstra et al., 2015), by differences in soil types and microbial communities (Williams and Rice, 2007, Collins et al., 2008, Garcia-

Palacios et al., 2016, Bhattacharyya et al., 2022), or by differences in litter quality (Melillo et al. 1982, Brovkin et al., 2012). With our approach, we were able to model the direct effect of temperature and moisture on turnover rates while accounting for site-level differences (e.g., microbial or plant communities) in how sensitive turnover is to soil moisture and temperature. This was done through data-assimilation estimation of $Q_{10}$ (temperature sensitivity of C turnover) and *mscut* (soil moisture sensitivity of C turnover) parameters (Fig. A5). At four of the six grasslands, both moisture and temperature had strong

effects on C turnover during the growing season (Fig. 3a,b,d,e), which corresponds to well-known moisture and temperature controls on microbial activity (Bell et al., 2008). However, in the mesic tallgrass prairie (KNZ) and the cooler shortgrass prairie (CPER), we found moisture limitation on C turnover was minimal (Fig. 3c,f). At KNZ, this was likely due to relatively high soil moisture levels throughout the growing season (Table B6). In conjunction with soil temperatures optimal for microbial activity, this resulted in high C turnover rates throughout the growing season at KNZ (Fig. 3f) and low overall

ecosystem C residence times.

    Alternately, soils at CPER are coarse (Table B1) and become very dry during later months of the growing season, yet C turnover was not limited within the model by soil moisture. The lack of sensitivity of C turnover to soil moisture may be due to microbial communities adjusted to low soil moisture conditions at the site. The *mscut* parameter in TECO represents the soil moisture level at which C turnover – and by inference, soil microbial activity – in the system becomes limited. Because

we were able to use daily soil $CO_2$ fluxes that were directly linked with soil temperature and soil moisture data for at least two years at each site, our estimates of *mscut* (and $Q_{10}$) parameters were remarkably well constrained (Fig. A4; Table B2).

The estimate of the *mscut* parameter at CPER was 7%, the lowest of all six sites. This means that when volumetric soil moisture is above 7%, microbial activity is not restricted by soil moisture in the model. Soils at CPER during the growing season (June-Sept) were the driest of all the sites, having an average soil volumetric water content (VWC) of 11% (Table B6). Additionally, the site having the second driest soils (12% VWC – SBK) also had a low *mscut* estimate (Fig. A4). This raises the interesting possibility that ecosystems with drier soils have microbial communities adapted to low water conditions, which would result in C turnover rates persisting even in relatively xeric conditions. If it is the case that C turnover is less responsive to altered soil moisture, this could result in mismatches between responses of C inputs versus outputs under altered precipitation regimes, since NPP has been shown to be highly sensitive to precipitation in more arid ecosystems (Huxman et al., 2004, Sala et al., 2012, Maurer et al., 2020).

Lignin and cellulose contents of litter have been shown to be important drivers of turnover rates (Adair et al. 2008), which can lead to differences in turnover rates of $C_3$ versus $C_4$ vegetation (Brovkin et al. 2012). As such, we would expect that grassland sites dominated by $C_3$ grasses would have shorter ecosystem C residence times compared with grasslands dominated by $C_4$ vegetation. Indeed, within the HPG and CPER sites where $C_3$ abundance is greatest among the six sites, turnover rates for leaf ($c_1$) and fast soil organic matter ($c_3$) were estimated to be relatively fast (Fig. A2, Table B2). This likely reflects the high cellulose and nitrogen in aboveground plant material at these sites (Blumenthal et al., 2020). However, estimates of ecosystem C residence time at HPG and CPER were not particularly low (Table 3). At HPG, this was largely due to a slow estimated root turnover rate, which resulted in an increased estimate of ecosystem C residence time (positive $c_2$ bar in Fig. 3j). At CPER, the temperature scalar strongly limited turnover rates (Fig. 3c). The cause of this may be a combination of the colder temperatures and a low $Q_{10}$ estimate at CPER. Only the HPG is cooler, yet the $Q_{10}$ estimate at HPG is much greater than at CPER (Fig. A5, Table B2). This raises the interesting possibility that the activity of decomposer communities at the shortgrass site may be less sensitive to temperature than other microbial communities, although additional inquiries are necessary to assess potential mechanisms. It could also be that C in CPER is older, and thus less susceptible to losses via decomposition (Conant et al., 2011). This idea fits with findings from findings of Liski et al., (1999) showing that decomposition rates of old soil C are relatively temperature insensitive, which aligns with the low $Q_{10}$ value estimated for CPER. Ultimately, we did not find strong effects of the abundance of vegetation functional groups on C turnover rates in this study (Table B5). This does not preclude the importance of functional composition on C cycling, we feel it simply suggests that there are other factors that are outweighing these effects on C dynamics.

Turnover rates of more recalcitrant pools of soil carbon have the potential to have strong influences on ecosystem C residence times, as has been shown using a long-term cross-site decomposition experiment (Harmon et al., 2009). Turnover rates of more recalcitrant pools are represented in this study using the $c_5$ and $c_6$ parameters, which represent the amount of C lost from the slow and passive C pools each day. It is important to note that these turnover estimates represent the inherent turnover rates without the effects of moisture and temperature. The estimates for the $c_5$ parameter were well constrained

across all sites by the data assimilation process (Figure A2) and turnover rate estimates were relatively slow, ranging from 8.7 years at HPG to 21.4 years at CPER. Indeed, the turnover rate of the slow pool at CPER caused ecosystem C residence time to be substantially greater at that site (Fig. 3i).

The potential distribution of the $c_6$ parameter resulting from the data assimilation process was quite broad for each site (Fig. A2), which is likely a big part of the uncertainty present within our estimated ecosystem C residence times and the resultant uncertainty around the estimate of C capacity (grey bars in Fig. 5). For example, the MLE of the $c6$ parameter for the black grama grassland was 1.55E-05 g C $\cdot$ g C$^{-1}$ $\cdot$ day$^{-1}$ (Table B2), which translates to a baseline turnover rate of 177 years for the passive C pool. Yet, the lower bound of the 95% confidence interval for the $c6$ parameter at the same site results in a baseline turnover rate of 1940 years. Differences in passive C turnover rates can have substantial effects on ecosystem C residence times, as shown in our sensitivity analysis (Fig. 4). In terms of our example above, going from passive C turnover rates of 177 to 1940 years causes ecosystem C residence time to go from 27 to 62 years. In this example, the magnitude of change of the ecosystem C residence time is less than that of the passive C turnover rates because ecosystem C residence time incorporates turnover of many other C pools, and not all C molecules end up in recalcitrant C pools. The overall effect on C saturation is substantial and important. We would like to note that variation in the passive and slow C turnover rates was a major component of the uncertainty in C capacity and saturation estimates (Fig. 5). This observation highlights the importance of these recalcitrant C pools for limiting losses of C from ecosystems.

**4.3 Patterns of C saturation**

Three of the grasslands we assessed had large gaps between present C and C capacity (CPER, HPG, and HAR; Fig. 5). Similar to forests acting as long-term C sinks during recovery from clear cutting regimes (Pan et al. 2011), it is possible that these cooler and/or wetter grassland ecosystems will act as C sinks due to long-term agricultural or other land-management legacies (Smith, 2014). These systems may be buffered against C losses if environmental changes occur, at least in the short-term. Alternately, the two hot and dry ecosystems showed high C saturation levels (Fig. 5), which corresponds with previous work at the SBK site showing this system is often a C source (Petrie et al., 2015), although long-term SOC data from this site indicate no net change in total soil C over time (Hou et al. 2021). High C saturation in these systems may also lead to C losses in the future, especially if global changes chronically reduce either NPP or C residence times. Short term effects on NPP or C residence times, such as those imposed by drought, may not have as strong effects on soil C because they do not permanently modify the capacity to store C of these systems, which may explain a lack of response of soil C in drought experiments (Holguin et al., 2022).

KNZ had short C residence time, but it also had the highest NPP estimate (Table 2), which should have resulted in a high C capacity. Yet, this system is burned annually in the spring, reflecting common management practices in this region (Knapp et al., 1998; Freckleton, 2004). Burning minimizes the amount of aboveground tissue that is incorporated into the soil due to

volatilization of C to the atmosphere (Seastedt, 1988), although some C is deposited as pyrogenic C (Soong and Contrufo, 2015). Despite these annual losses, C capacity is still relatively close to present C, perhaps due to increased root production under frequent fire regimes (Johnson and Matchett, 2001). This may be one reason that research in this ecosystem has found that soil C is resistant to altered environmental conditions despite frequent fire (Wilcox et al., 2016), which has been predicted to reduce soil C through time due to losses through volatization (Ojima et al. 1994).

Our estimates of soil C extended from 0-20 cm in the soil, which presents two potential limitation in this study. First, since our measurements of soil C were from 0-10 cm in the soil, it was necessary to extrapolate this C to 20 cm using depth-soil C relationships from each site. This was important to match depths with all the other measurements used in this study, but the extrapolation process introduces additional sources of error. Additionally, our estimates of both C capacity and present C did not include measurements of C below 20 cm in the soil. Deeper C represents an additional store of C in many ecosystems

and may stabilize C in ecosystems where C inputs by roots at depth are frequent, such as savanna and shrubland ecosystems. Additionally, soil microbial communities differ markedly in deeper soil layers than shallow layers (Fierer et al., 2003), so turnover rates across these systems may be quite different for deeper soils. Yet in many grasslands, C fluxes aboveground and in shallow soil layers are more likely to respond under global change scenarios than deeper soils because: (1) the proportion of root production of soil organic C and microbial activity are typically greatest within shallow soil layers

(Jackson et al., 1996, Jobággy and Jackson, 2000, Blume et al., 2002, Taylor et al., 2002), and (2) altered air temperature is more likely to impact soil temperatures in surface soils versus deeper soils. Another potential limitation of our study here is that we did not incorporate uncertainty among plot-level measurements of CO2 surface flux (Fig. A9). As such, our estimates apply to the site-level since we performed data-model fusion using site-level averages of our measurements. However, it would be interesting to examine how these ecological processes and properties varied at smaller spatial scales

(*i.e.*, across plots).

## 5 Conclusions

Here we used a recently developed metric, carbon capacity, to assess potential future trajectories of ecosystem C across six grasslands in the US Great Plains, and to identify grasslands that may be vulnerable to C loss under future global change scenarios. We showed that hot and dry grasslands had C contents greater than their C capacity, suggesting future C loss in

these systems, especially if environmental conditions continue to change. As arid ecosystems have been shown to be key components of the global C cycle due to their broad spatial extent (Poulter et al., 2014), understanding how NPP and ecosystem carbon residence times respond to alterations in environmental conditions in these ecosystems is vital for assessing future global C budgets. Additionally, the effect of frequent burning on C saturation suggests that land management practices that remove aboveground biomass may result in reduced capacity for these systems to be C sinks into

the future. However, the effects of disturbances such as fire are complex and often are critical to maintaining ecosystem structure so holistic consideration of all effects is important for management decisions. Because anthropogenic and climate

effects on ecosystems are global and ubiquitous, considerations of how land management and environmental impacts interact to control ecosystem functioning are critical for land management and policy decisions related to C sequestration.

**Appendices:**

Appendix A. Supplemental Figures A1-A10

Appendix B. Supplemental Tables B1-B6

Appendix C. Supplemental text describing data collection and cleaning methods

Appendix D. Model evaluation text, Table D1, Figures D1-D2

Appendix E. Land use history of study sites





**Appendix A. Supplemental figures A1-A8**

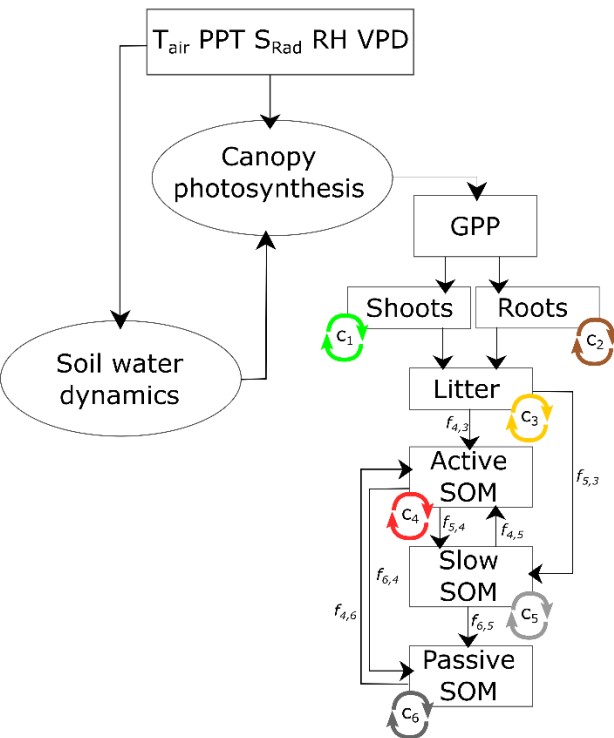

Figure A1. Schematic of the submodels making up the Terrestrial Ecosystem model. Canopy photosynthesis is determined by hourly meteorological information through direct impacts (e.g., vapor pressure deficit) and alterations to soil environmental conditions (e.g., soil moisture and temperature). GPP is estimated from the canopy photosynthesis model. Carbon from the canopy model is transferred to different components of vegetation, which is summed over an annual time step to estimate NPP (GPP – autotrophic respiration). See Weng and Luo (2008) for additional details of the model. The soil C model incorporates turnover rates for aboveground vegetation ($c_1$), roots ($c_2$), litter ($c_3$), active SOM ($c_4$), slow SOM ($c_5$), and passive SOM ($c_6$). Turnover rates are modified by an environmental scalar ($\tau$) at each time step. The proportion of C transferred between pools are controlled by the $f_{i,j}$ parameters, which represent the proportion of C turnover transferred from C pool $j$ to pool $I$, with the pool numbers corresponding to the subscripts by the C turnover parameters.

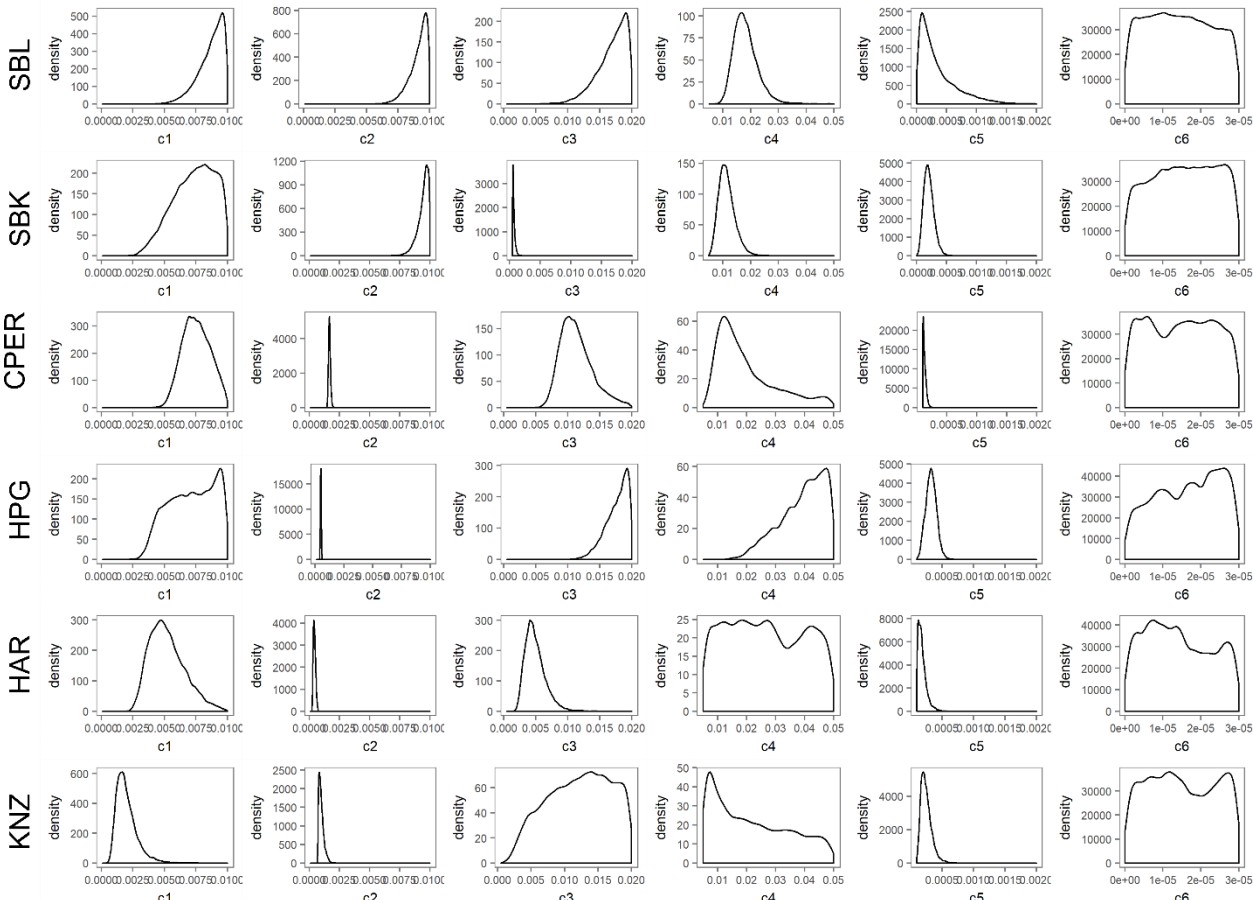

**Figure A2.** Density plots showing distribution of estimated carbon turnover parameters (columns) using data assimilation techniques with a 6 pool carbon model and 4-5 data sets describing carbon pools and fluxes at each of the six sites (rows). Densities reflect 4 chains of 360k simulations each, with the first 20k simulations removed. SBL= Blue grama dominated site at the Sevilleta National Wildlife Refuge; SBK= Black grama dominated site at the Sevilleta National Wildlife Refuge; CPER=central plains experimental range; HPG=High Plains Grassland Research Station; HAR=Hays Agricultural Research Station; KNZ=Konza Prairie Biological Station. All carbon turnover parameters are in units of gC lost $gC^{-1}$ $day^{-1}$: c1=leaf turnover; c2=root turnover; c3=litter turnover; c4=fast SOM turnover; c5=slow SOM turnover; c6=passive SOM turnover.

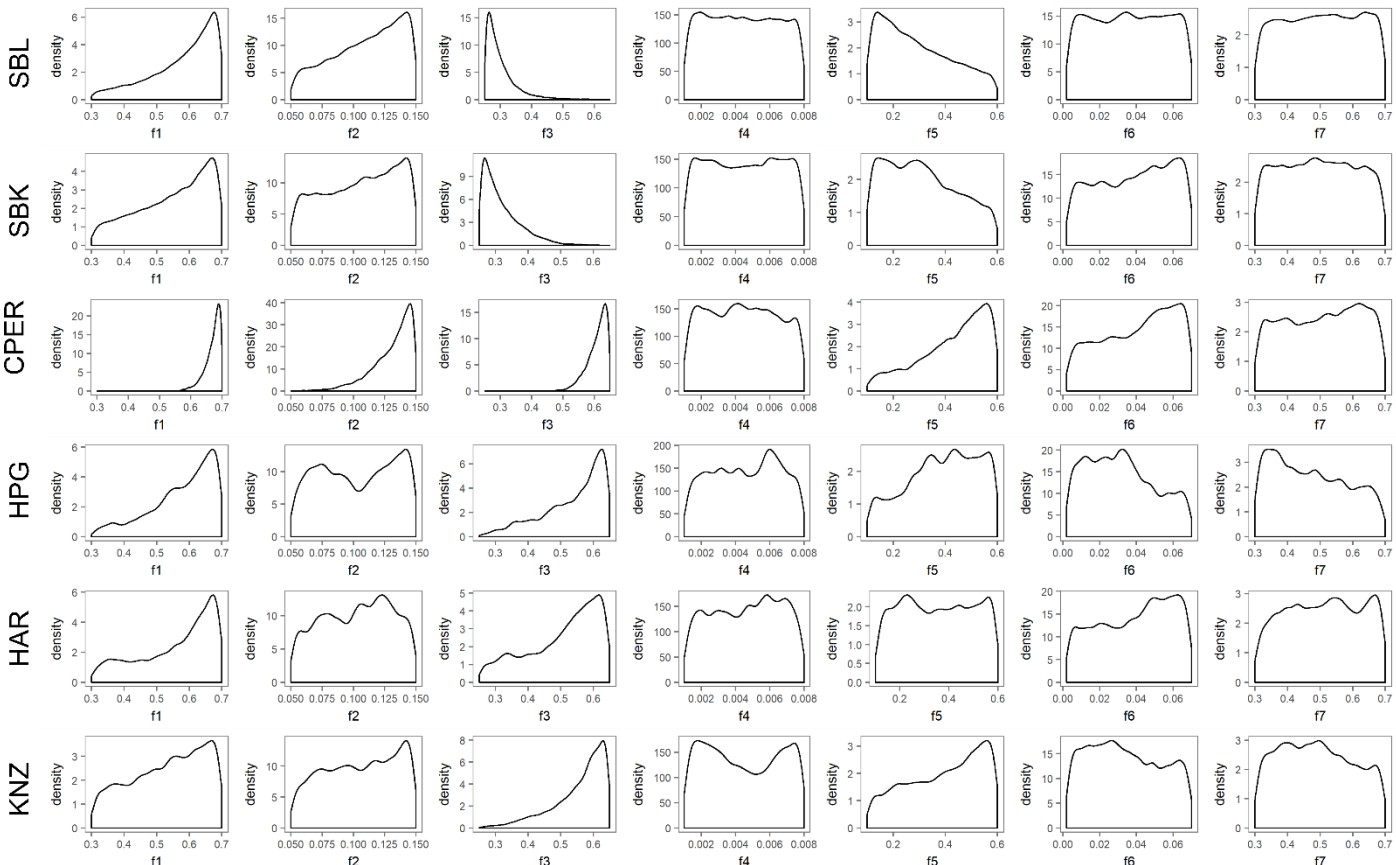

**Figure A3. Density plots showing distribution of estimated carbon transfer parameters (columns) using data assimilation techniques with a 6 pool carbon model and 4-5 data sets describing carbon pools and fluxes at each of the six sites (rows). Densities reflect 4 chains of 360k simulations each, with the first 20k simulations removed. SBL= Blue grama dominated site at the Sevilleta National Wildlife Refuge; SBK= Black grama dominated site at the Sevilleta National Wildlife Refuge; CPER=central plains experimental range; HPG=High Plains Grassland Research Station; HAR=Hays Agricultural Research Station; KNZ=Konza Prairie Biological Station. Transfer parameters (fx,y) dictate the proportion of C turnover in pool y transferring to pool x: f1=f43; f2=f53; f3=f54; f4=f64; f5=f45; f6=f65; f7=f46.**

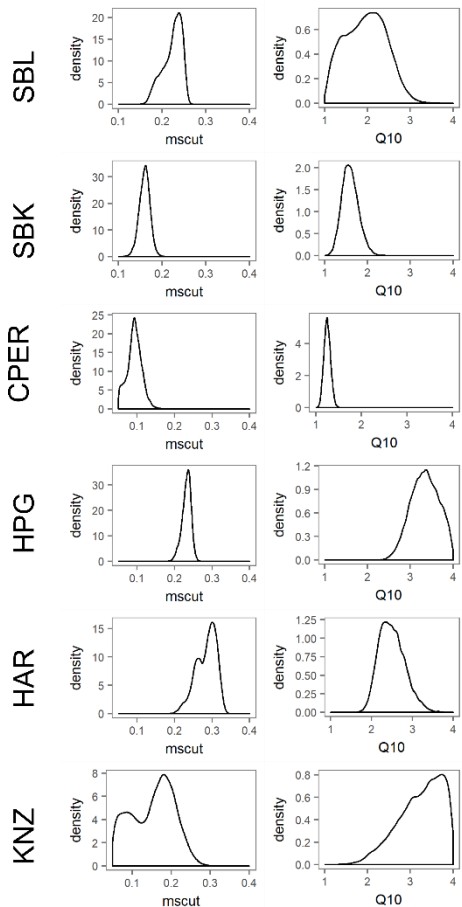

**Figure A4. Density plots showing distribution of environmental scaling parameters (columns) dictating decomposition rates. We used data assimilation techniques with a 6 pool carbon model and 4-5 data sets describing carbon pools and fluxes at each of the six sites (rows). Densities reflect 4 chains of 360k simulations each, with the first 20k simulations removed. SBL= Blue grama dominated site at the Sevilleta National Wildlife Refuge; SBK= Black grama dominated site at the Sevilleta National Wildlife Refuge; CPER=central plains experimental range; HPG=High Plains Grassland Research Station; HAR=Hays Agricultural Research Station; KNZ=Konza Prairie Biological Station. Mscut is the soil moisture level at which decomposition rates begin to become water limited; Q10 is the temperature sensitivity of decomposition.**

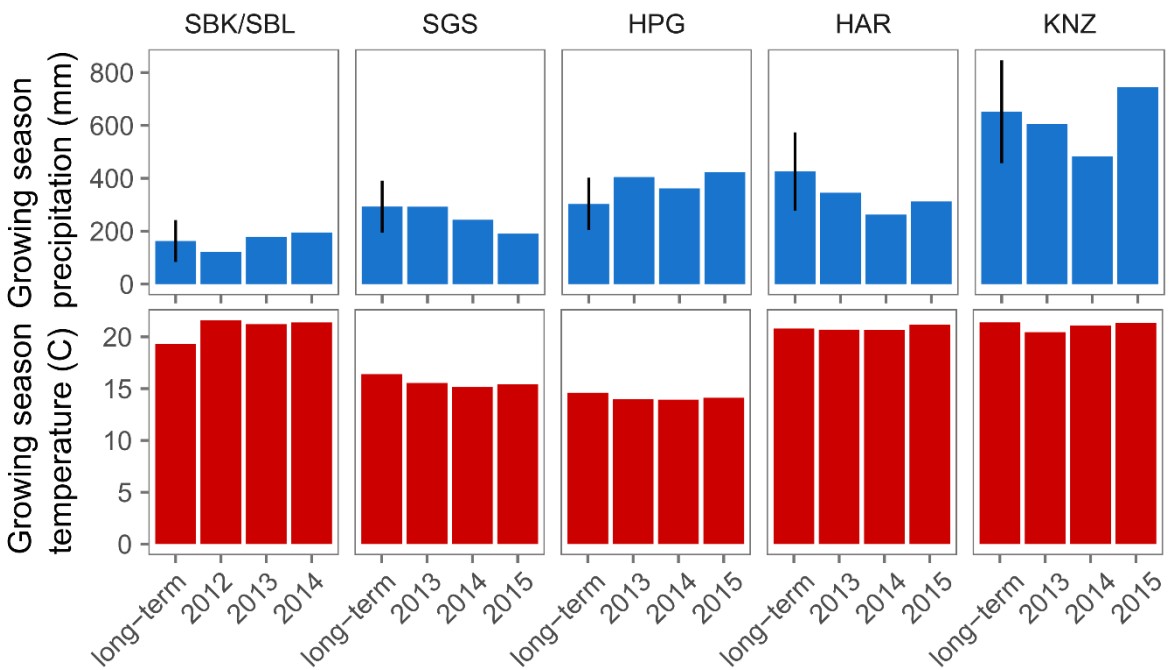

**Figure A5. Precipitation and air temperature during the growing season (Apr-Oct for SBK and SBL, Apr-Sept for all other sites) during the three focal years at each site compared with the long-term mean precipitation and air temperature. Averages represent the period of 1982-2012 obtained from Knapp et al. 2015, and error bars represent 1 standard deviation from the mean.**

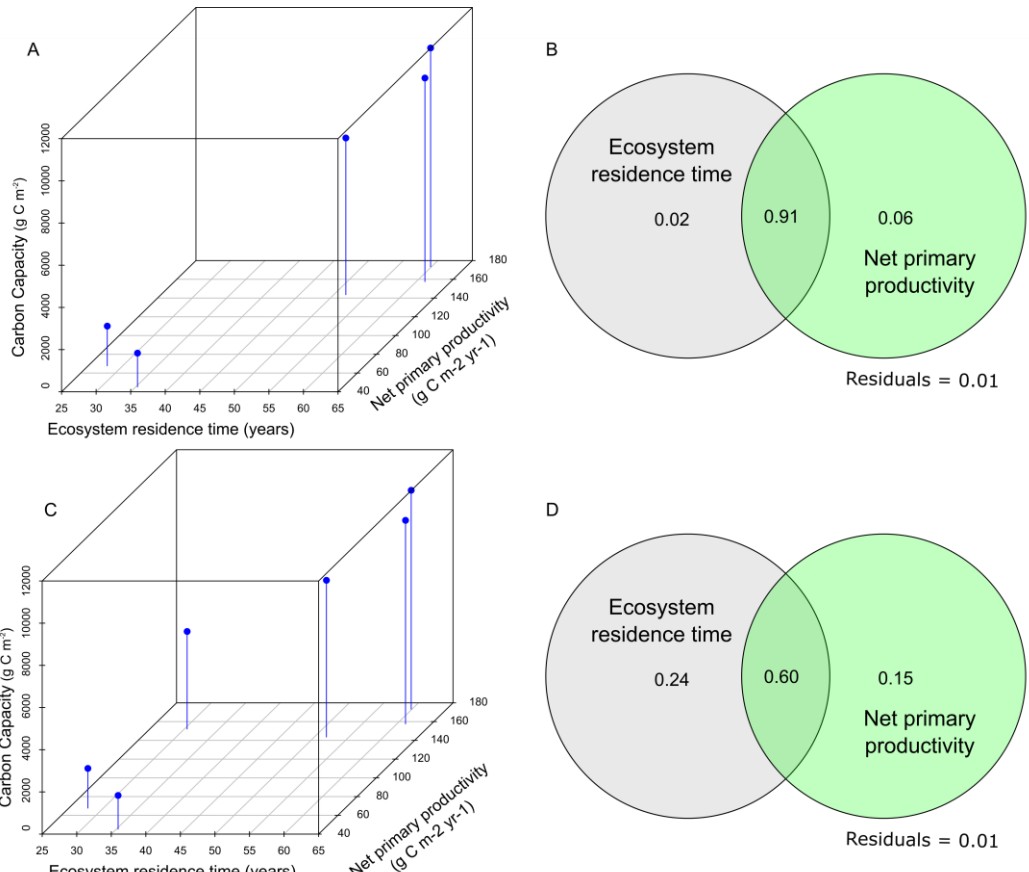

Figure A6. Three-dimensional plots (A,C) and variance partitioning results (B,D) showing the importance of ecosystem residence time and net primary productivity in making up the systems carbon capacity for our six focal ecosystems. A and B panels show results without KNZ, C and D include KNZ. In (B,D), numbers within the circles represent the amount of cross-site variance in carbon capacity explained solely by ecosystem residence time or net primary productivity. The number in the intersection represent variance explained jointly by both components.

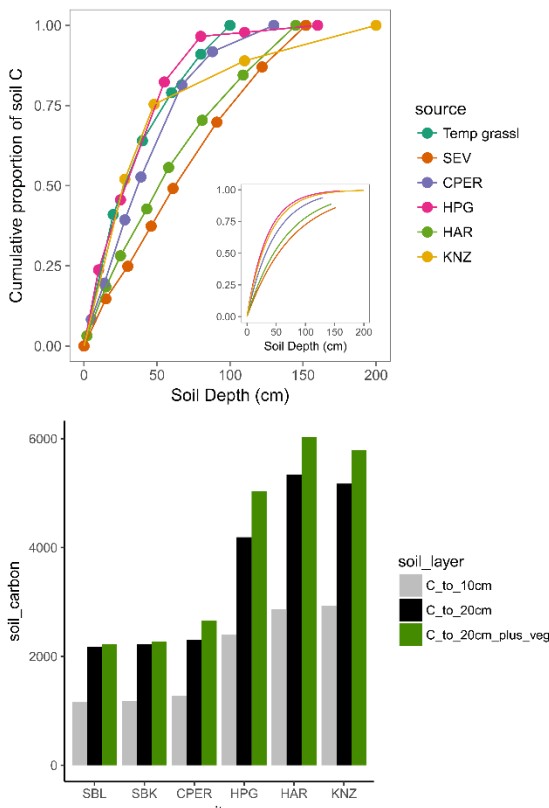

Figure A7. Top panel: Cumulative proportion of soil C along depth profile (main) and beta distribution of proportion by depth (inset). Data were obtained from the International Soil Carbon Network (see table B4) from areas close to study sites and having similar cover types and management regimes. In main panel, we included estimated proportions for temperate grasslands (Temp grass) from Jobbágy and Jackson (2000). Bottom panel: soil C measured at each study site from 0-10 cm and soil C estimates from 0-20 cm based on relationships in top panel. Estimates were obtained using: $C_{20}=C_{10}/C_{P10} \cdot C_{P20}$, where $C_{10}$ is the soil C measured in the top 10 cm, $C_{P10}$ is the proportion of C in the top 10 cm based on the beta regression, and $C_{P20}$ is the proportion of C in the top 20 cm. Green bars represent soil C estimates from 0-20 cm plus vegetation C.

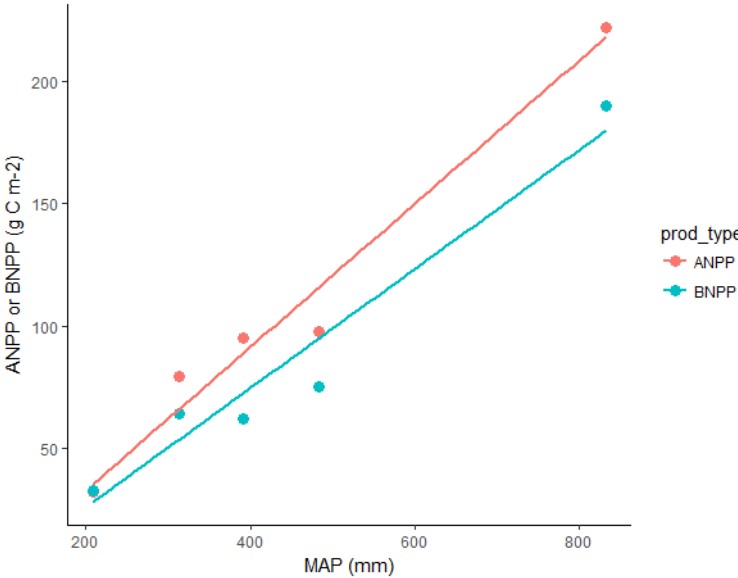

**Figure A8. Average ANPP and BNPP simulation output across mean annual precipitation of our six grassland sites.**



**Figure A9. Soil surface CO2 flux estimates at midday for all six sites throughout the study period, used within the data assimilation process to compare with model output. Units are µmol m⁻² s⁻¹. Average daily standard error among plots are as follows: sevblk: 0.075, sevblu: 0.14, cper: 0.10, hpg: 0.49, hys: 0.57, knz: 0.47, although plot to plot variation is not included in data assimilation procedure.**

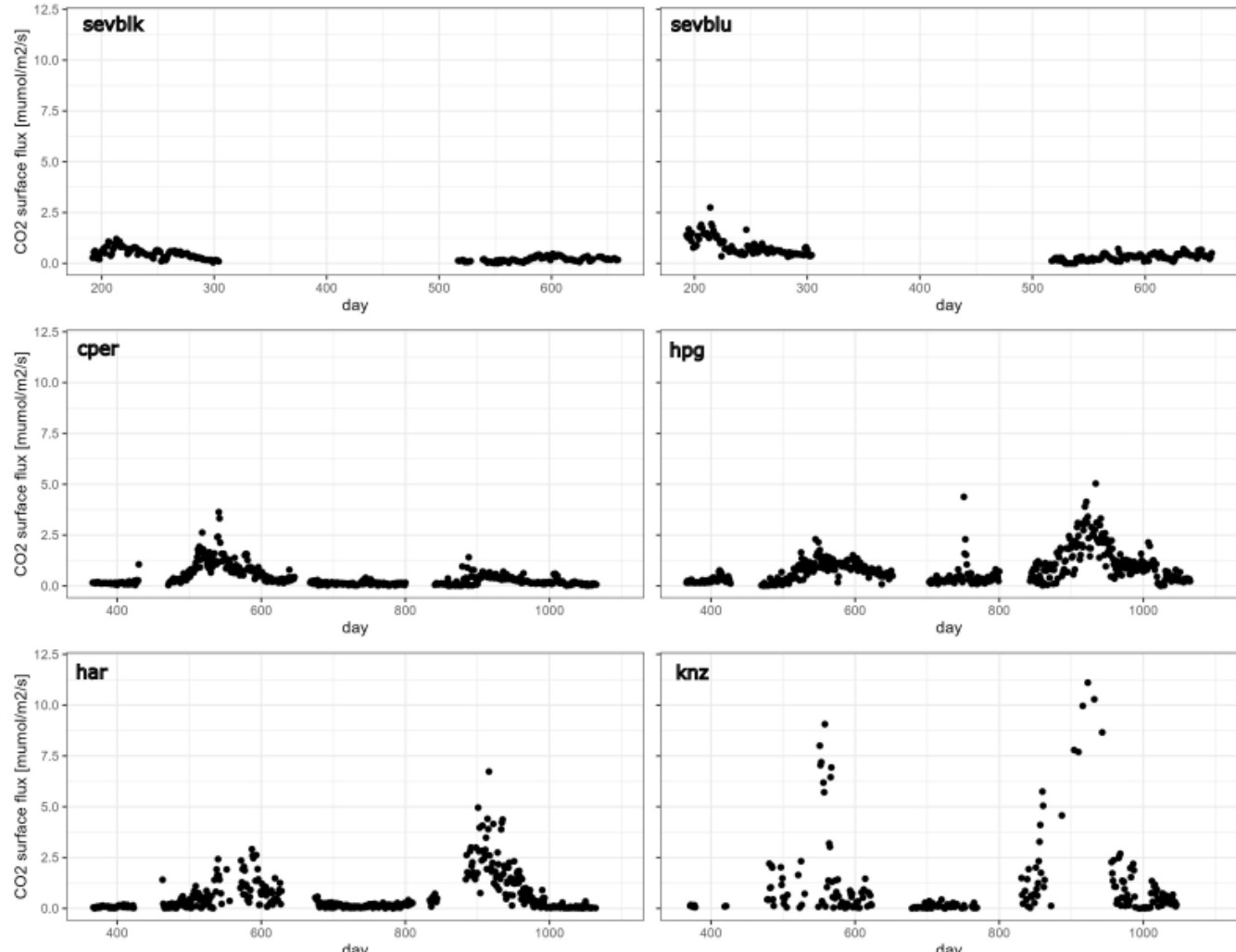

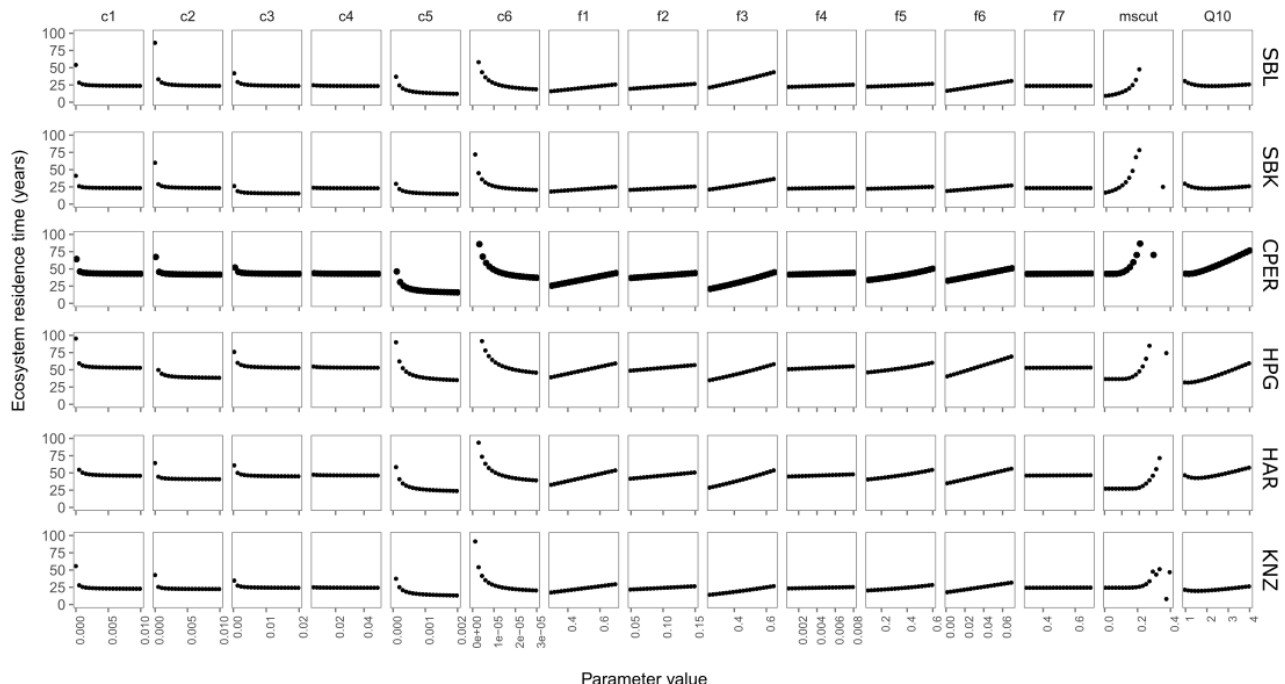

**Figure A10. Results from sensitivity analysis where ecosystem C residence time was calculated when altering one parameter value at a time. Rows of panels correspond to different sites with: SBL= Blue grama dominated site at the Sevilleta National Wildlife Refuge; SBK= Black grama dominated site at the Sevilleta National Wildlife Refuge; CPER=central plains experimental range; HPG=High Plains Grassland Research Station; HAR=Hays Agricultural Research Station; KNZ=Konza Prairie Biological Station. Ecosystem carbon residence time was often very high at very extreme parameter values so the y axis was set for clarity. Transfer parameters (f1-7) are as in Fig. 3.**




**Appendix B. Supplemental tables B1-B6**

| Param. | Default | Lower | Upper | Description | Units |
|---|---|---|---|---|---|
| c1 | 1.00E-03 | 1.00E-04 | 1.00E-02 | The proportion of leaf C turning over each day | gC gC-1 day-1 |
| c2 | 9.00E-03 | 1.00E-04 | 1.00E-02 | The proportion of root C turning over each day | gC gC-1 day-1 |
| c3 | 9.00E-03 | 5.00E-04 | 2.00E-02 | The proportion of litter C turning over each day | gC gC-1 day-1 |
| c4 | 1.50E-02 | 5.00E-03 | 5.00E-02 | Proportion of fast SOM turning over each day | gC gC-1 day-1 |
| c5 | 6.00E-04 | 1.00E-05 | 2.00E-03 | Proportion of slow SOM turning over each day | gC gC-1 day-1 |
| c6 | 2.00E-05 | 1.00E-08 | 3.00E-05 | Proportion of passive SOM turning over each day | gC gC-1 day-1 |
| f43 | 2.50E-01 | 3.00E-01 | 7.00E-01 | Proportion of litter C turnover going to fast SOM pool | - |
| f53 | 1.00E-01 | 5.00E-02 | 1.50E-01 | Proportion of litter C turnover going to slow SOM pool | - |
| f54 | 5.00E-01 | 2.50E-01 | 6.50E-01 | Proportion of fast SOM turnover going to slow SOM pool | - |
| f64 | 4.00E-03 | 1.00E-03 | 8.00E-03 | Proportion of litter C turnover going to passive SOM pool | - |
| f45 | 4.20E-01 | 1.00E-01 | 6.00E-01 | Proportion of slow SOM turnover going to fast SOM pool | - |
| f65 | 5.00E-02 | 2.00E-03 | 7.00E-02 | Proportion of slow C turnover going to passive SOM pool | - |
| f46 | 4.50E-01 | 3.00E-01 | 7.00E-01 | Proportion of passive C turnover going to fast SOM pool | - |
| Q10 | 2.20E+00 | 1.00E+00 | 4.00E+00 | Temperature sensitivity of decomposition | - |
| mscut | 2.00E-01 | 1.00E-02 | 4.00E-01 | Soil moisture level at which decomposition starts to become water limited | - |

**Constraints and descriptions of carbon cycling and environmental scaling parameters used in data assimilation.**



**Table B2. C cycling and environmental scaling parameters estimated via data assimilation with a six pool C model and 4-5 C pool and flux data sets for each of six grassland sites. 95% confidence intervals (CLl and cLu) are estimated from normal, log-normal, or Weibull distributions depending on the magnitude and direction of skew. Gelman-Rubin statistics (G-R) indicate convergence among independent chains.**

| | SBL | | | | SBK | | | |
|---|---|---|---|---|---|---|---|---|
| Param | MLE | CLl | cLu | G-R | MLE | CLl | cLu | G-R |
| c1 | 0.009107 | 6.75E-03 | 1.00E-02 | 1.0 | 0.007977 | 0.004471 | 0.009999 | 1.0 |
| c2 | 0.009405 | 7.70E-03 | 1.00E-02 | 1.0 | 0.00963 | 0.008492 | 0.01 | 1.0 |
| c3 | 0.017967 | 1.24E-02 | 2.00E-02 | 1.0 | 0.00068 | 0.0005 | 0.001112 | 1.0 |
| c4 | 0.017657 | 1.06E-02 | 2.67E-02 | 1.0 | 0.01101 | 0.006237 | 0.0172 | 1.0 |
| c5 | 0.000232 | 1.00E-05 | 1.02E-03 | 1.0 | 0.000188 | 5.61E-05 | 0.000374 | 1.0 |
| c6 | 1.46E-05 | 8.02E-07 | 2.87E-05 | 1.0 | 1.55E-05 | 1.41E-06 | 2.93E-05 | 1.0 |
| f43 | 0.618122 | 3.74E-01 | 7.00E-01 | 1.0 | 0.593098 | 0.347119 | 0.699949 | 1.0 |
| f53 | 0.110032 | 6.02E-02 | 1.50E-01 | 1.0 | 0.1055 | 0.056978 | 0.149515 | 1.0 |
| f54 | 0.298433 | 2.50E-01 | 4.10E-01 | 1.0 | 0.312915 | 0.250004 | 0.435435 | 1.0 |
| f64 | 0.004441 | 1.16E-03 | 7.71E-03 | 1.0 | 0.004506 | 0.001239 | 0.007792 | 1.0 |
| f45 | 0.262651 | 1.00E-01 | 5.45E-01 | 1.0 | 0.31254 | 0.103977 | 0.552567 | 1.0 |
| f65 | 0.036233 | 4.11E-03 | 6.77E-02 | 1.0 | 0.038417 | 0.005276 | 0.068779 | 1.0 |
| f46 | 0.504545 | 3.17E-01 | 6.90E-01 | 1.0 | 0.497965 | 0.312594 | 0.681608 | 1.0 |
| mscut | 0.223073 | 1.78E-01 | 2.56E-01 | 1.0 | 0.159795 | 0.135553 | 0.18398 | 1.0 |
| Q10 | 1.895981 | 1.07E+00 | 2.76E+00 | 1.0 | 1.597074 | 1.231482 | 1.986987 | 1.0 |

| | CPER | | | | HPG | | | |
|---|---|---|---|---|---|---|---|---|
| Param | MLE | CLl | cLu | G-R | MLE | CLl | cLu | G-R |
| c1 | 0.00746 | 0.005609 | 0.00966 | 1.0 | 0.007872 | 0.004192 | 0.009959 | 1.1 |
| c2 | 0.001675 | 0.001518 | 0.00184 | 1.0 | 0.000488 | 0.000417 | 0.000562 | 1.0 |
| c3 | 0.011091 | 0.006979 | 0.016717 | 1.0 | 0.018395 | 0.0143 | 0.019997 | 1.0 |
| c4 | 0.017479 | 0.006234 | 0.041909 | 1.0 | 0.042184 | 0.024925 | 0.049978 | 1.1 |
| c5 | 0.000128 | 0.0001 | 0.000181 | 1.0 | 0.000314 | 0.000164 | 0.000482 | 1.0 |
| c6 | 1.49E-05 | 7.57E-07 | 2.86E-05 | 1.1 | 1.64E-05 | 1.97E-06 | 2.92E-05 | 1.2 |
| f43 | 0.68024 | 0.618193 | 0.699996 | 1.0 | 0.576301 | 0.377749 | 0.699959 | 1.1 |
| f53 | 0.137347 | 0.099531 | 0.149997 | 1.0 | 0.102849 | 0.057045 | 0.14747 | 1.3 |
| f54 | 0.62252 | 0.546666 | 0.649987 | 1.0 | 0.574702 | 0.35944 | 0.649937 | 1.1 |
| f64 | 0.004421 | 0.001187 | 0.007682 | 1.1 | 0.004583 | 0.001361 | 0.00768 | 1.2 |
| f45 | 0.427899 | 0.176967 | 0.599756 | 1.0 | 0.388533 | 0.150328 | 0.587869 | 1.2 |
| f65 | 0.040839 | 0.007248 | 0.069772 | 1.1 | 0.032149 | 0.002948 | 0.060738 | 1.1 |
| f46 | 0.506596 | 0.316463 | 0.687456 | 1.0 | 0.473907 | 0.305835 | 0.667426 | 1.2 |
| mscut | 0.073406 | 0.014973 | 0.125796 | 1.0 | 0.231396 | 0.205847 | 0.253547 | 1.0 |
| Q10 | 1.248167 | 1.10994 | 1.393126 | 1.0 | 3.482896 | 2.773369 | 3.950379 | 1.0 |

| Param | HAR | | | | KNZ | | | |
|---|---|---|---|---|---|---|---|---|
| | MLE | CLl | cLu | G-R | MLE | CLl | cLu | G-R |
| c1 | 0.005025 | 0.002799 | 0.007883 | 1.0 | 0.001854 | 0.000722 | 0.00367 | 1.0 |
| c2 | 0.000414 | 0.000257 | 0.0006 | 1.0 | 0.000967 | 0.000699 | 0.001413 | 1.0 |
| c3 | 0.004664 | 0.002313 | 0.007928 | 1.0 | 0.013651 | 0.004197 | 0.019902 | 1.0 |
| c4 | 0.026455 | 0.006518 | 0.047266 | 1.0 | 0.021989 | 0.005002 | 0.045437 | 1.0 |
| c5 | 0.000171 | 0.0001 | 0.000308 | 1.0 | 0.000233 | 0.000106 | 0.00041 | 1.0 |
| c6 | 1.41E-05 | 9.88E-07 | 2.88E-05 | 1.0 | 1.50E-05 | 1.46E-06 | 2.94E-05 | 1.0 |
| f43 | 0.561531 | 0.34928 | 0.699978 | 1.0 | 0.531489 | 0.347243 | 0.696558 | 1.1 |
| f53 | 0.102973 | 0.057002 | 0.146251 | 1.1 | 0.104992 | 0.05725 | 0.148682 | 1.1 |
| f54 | 0.54796 | 0.303471 | 0.649941 | 1.3 | 0.586921 | 0.387489 | 0.649992 | 1.0 |
| f64 | 0.004599 | 0.001247 | 0.007561 | 1.2 | 0.004439 | 0.001255 | 0.007846 | 1.0 |
| f45 | 0.354072 | 0.125197 | 0.577835 | 1.3 | 0.39581 | 0.156163 | 0.599482 | 1.1 |
| f65 | 0.03924 | 0.006196 | 0.069346 | 1.1 | 0.034188 | 0.004299 | 0.066817 | 1.1 |
| f46 | 0.509573 | 0.327535 | 0.691565 | 1.0 | 0.490831 | 0.314648 | 0.683017 | 1.1 |
| mscut | 0.283857 | 0.231203 | 0.328458 | 1.0 | 0.114386 | 0.013291 | 0.232586 | 1.1 |
| Q10 | 2.503186 | 1.929534 | 3.128982 | 1.0 | 3.445054 | 2.277028 | 3.999685 | 1.0 |





**Table B3. Cross correlations between Markov Chain Monte Carlo output of different model parameters during data assimilation.**

| Site | param | c1 | c2 | c3 | c4 | c5 | c6 | f43 | f53 | f54 | f64 | f45 | f65 | f46 | mscut | Q10 |
|---|---|---|---|---|---|---|---|---|---|---|---|---|---|---|---|---|
| SBL | c1 | 1 | 0.06 | 0.05 | 0.14 | 0.11 | 0.02 | -0.04 | -0.04 | 0.04 | 0.01 | -0.02 | 0.02 | -0.01 | -0.09 | -0.17 |
| SBL | c2 | 0.06 | 1 | 0.04 | 0.13 | 0.02 | 0 | -0.04 | -0.02 | 0 | 0 | 0.01 | 0.01 | 0 | 0.05 | -0.07 |
| SBL | c3 | 0.05 | 0.04 | 1 | 0.1 | -0.02 | 0.03 | 0.11 | -0.01 | -0.02 | 0.01 | 0.01 | 0.01 | -0.02 | 0.01 | -0.07 |
| SBL | c4 | 0.14 | 0.13 | 0.1 | 1 | 0.47 | 0.02 | 0 | -0.05 | 0.17 | 0.02 | -0.05 | 0.02 | -0.01 | -0.18 | -0.43 |
| SBL | c5 | 0.11 | 0.02 | -0.02 | 0.47 | 1 | -0.03 | -0.03 | 0.02 | 0.41 | 0.03 | -0.21 | 0.06 | 0 | -0.79 | -0.86 |
| SBL | c6 | 0.02 | 0 | 0.03 | 0.02 | -0.03 | 1 | 0.01 | -0.01 | 0 | 0.05 | -0.02 | -0.02 | -0.05 | 0.02 | 0 |
| SBL | f43 | -0.04 | -0.04 | 0.11 | 0 | -0.03 | 0.01 | 1 | 0.01 | -0.01 | -0.02 | 0.04 | 0 | -0.04 | 0.1 | 0.18 |
| SBL | f53 | -0.04 | -0.02 | -0.01 | -0.05 | 0.02 | -0.01 | 0.01 | 1 | 0.01 | 0 | -0.04 | 0 | 0.01 | 0.04 | 0.09 |
| SBL | f54 | 0.04 | 0 | -0.02 | 0.17 | 0.41 | 0 | -0.01 | 0.01 | 1 | -0.03 | -0.08 | -0.01 | 0.03 | -0.22 | -0.21 |
| SBL | f64 | 0.01 | 0 | 0.01 | 0.02 | 0.03 | 0.05 | -0.02 | 0 | -0.03 | 1 | -0.04 | -0.01 | 0 | -0.04 | -0.04 |
| SBL | f45 | -0.02 | 0.01 | 0.01 | -0.05 | -0.21 | -0.02 | 0.04 | -0.04 | -0.08 | -0.04 | 1 | -0.03 | 0.03 | 0.25 | 0.28 |
| SBL | f65 | 0.02 | 0.01 | 0.01 | 0.02 | 0.06 | -0.02 | 0 | 0 | -0.01 | -0.01 | -0.03 | 1 | 0.02 | -0.03 | -0.02 |
| SBL | f46 | -0.01 | 0 | -0.02 | -0.01 | 0 | -0.05 | -0.04 | 0.01 | 0.03 | 0 | 0.03 | 0.02 | 1 | 0.01 | 0.01 |
| SBL | mscut | -0.09 | 0.05 | 0.01 | -0.18 | -0.79 | 0.02 | 0.1 | 0.04 | -0.22 | -0.04 | 0.25 | -0.03 | 0.01 | 1 | 0.87 |
| SBL | Q10 | -0.17 | -0.07 | -0.07 | -0.43 | -0.86 | 0 | 0.18 | 0.09 | -0.21 | -0.04 | 0.28 | -0.02 | 0.01 | 0.87 | 1 |
| | | | | | | | | | | | | | | | | |
| SBK | c1 | 1.00 | 0.05 | 0.01 | 0.09 | 0.11 | 0.01 | 0.03 | 0.03 | 0.02 | 0.04 | -0.01 | -0.02 | 0.04 | 0.00 | -0.16 |
| SBK | c2 | 0.05 | 1.00 | 0.05 | 0.15 | 0.17 | -0.01 | -0.01 | 0.00 | 0.02 | 0.01 | -0.01 | 0.00 | 0.01 | 0.07 | -0.21 |
| SBK | c3 | 0.01 | 0.05 | 1.00 | 0.05 | -0.17 | 0.01 | 0.08 | 0.06 | 0.01 | 0.00 | 0.01 | -0.02 | 0.00 | 0.05 | -0.16 |
| SBK | c4 | 0.09 | 0.15 | 0.05 | 1.00 | 0.52 | -0.01 | 0.00 | -0.02 | 0.06 | -0.01 | 0.04 | 0.02 | 0.02 | -0.01 | -0.44 |
| SBK | c5 | 0.11 | 0.17 | -0.17 | 0.52 | 1.00 | -0.07 | 0.09 | 0.01 | 0.27 | 0.01 | 0.11 | 0.06 | 0.01 | -0.31 | -0.75 |
| SBK | c6 | 0.01 | -0.01 | 0.01 | -0.01 | -0.07 | 1.00 | 0.04 | 0.04 | -0.02 | -0.06 | -0.05 | -0.01 | 0.03 | -0.02 | 0.00 |
| SBK | f43 | 0.03 | -0.01 | 0.08 | 0.00 | 0.09 | 0.04 | 1.00 | -0.05 | 0.09 | 0.00 | 0.01 | -0.08 | 0.06 | -0.02 | 0.00 |
| SBK | f53 | 0.03 | 0.00 | 0.06 | -0.02 | 0.01 | 0.04 | -0.05 | 1.00 | 0.02 | 0.04 | 0.04 | -0.08 | -0.10 | 0.00 | 0.02 |
| SBK | f54 | 0.02 | 0.02 | 0.01 | 0.06 | 0.27 | -0.02 | 0.09 | 0.02 | 1.00 | -0.02 | 0.02 | -0.03 | 0.01 | 0.03 | -0.02 |
| SBK | f64 | 0.04 | 0.01 | 0.00 | -0.01 | 0.01 | -0.06 | 0.00 | 0.04 | -0.02 | 1.00 | -0.02 | -0.01 | 0.00 | -0.01 | -0.01 |
| SBK | f45 | -0.01 | -0.01 | 0.01 | 0.04 | 0.11 | -0.05 | 0.01 | 0.04 | 0.02 | -0.02 | 1.00 | -0.06 | 0.00 | 0.06 | 0.07 |
| SBK | f65 | -0.02 | 0.00 | -0.02 | 0.02 | 0.06 | -0.01 | -0.08 | -0.08 | -0.03 | -0.01 | -0.06 | 1.00 | 0.01 | -0.01 | -0.02 |
| SBK | f46 | 0.04 | 0.01 | 0.00 | 0.02 | 0.01 | 0.03 | 0.06 | -0.10 | 0.01 | 0.00 | 0.00 | 0.01 | 1.00 | 0.00 | -0.01 |
| SBK | mscut | 0.00 | 0.07 | 0.05 | -0.01 | -0.31 | -0.02 | -0.02 | 0.00 | 0.03 | -0.01 | 0.06 | -0.01 | 0.00 | 1.00 | 0.53 |
| SBK | Q10 | -0.16 | -0.21 | -0.16 | -0.44 | -0.75 | 0.00 | 0.00 | 0.02 | -0.02 | -0.01 | 0.07 | -0.02 | -0.01 | 0.53 | 1.00 |
| | | | | | | | | | | | | | | | | |
| CPER | c1 | 1.00 | 0.18 | -0.05 | -0.08 | 0.02 | 0.04 | -0.01 | 0.04 | 0.05 | 0.01 | 0.02 | 0.02 | -0.02 | 0.17 | -0.16 |

| | | | | | | | | | | | | | | | |
|---|---|---|---|---|---|---|---|---|---|---|---|---|---|---|---|
| CPER | c2 | 0.18 | 1.00 | 0.09 | 0.00 | 0.06 | 0.02 | -0.02 | -0.03 | -0.03 | 0.01 | -0.02 | -0.02 | 0.00 | 0.50 | -0.47 |
| CPER | c3 | -0.05 | 0.09 | 1.00 | -0.30 | -0.01 | 0.07 | 0.08 | 0.02 | -0.06 | -0.07 | -0.05 | -0.04 | 0.00 | 0.08 | -0.02 |
| CPER | c4 | -0.08 | 0.00 | -0.30 | 1.00 | 0.01 | -0.04 | -0.03 | -0.09 | 0.08 | 0.06 | 0.00 | 0.01 | 0.10 | -0.03 | 0.01 |
| CPER | c5 | 0.02 | 0.06 | -0.01 | 0.01 | 1.00 | -0.02 | 0.09 | 0.09 | 0.17 | 0.00 | 0.18 | 0.05 | -0.03 | 0.03 | -0.13 |
| CPER | c6 | 0.04 | 0.02 | 0.07 | -0.04 | -0.02 | 1.00 | 0.03 | 0.05 | 0.05 | 0.02 | -0.04 | 0.05 | 0.03 | 0.00 | -0.02 |
| CPER | f43 | -0.01 | -0.02 | 0.08 | -0.03 | 0.09 | 0.03 | 1.00 | -0.08 | -0.11 | 0.03 | -0.06 | -0.02 | -0.02 | -0.02 | 0.08 |
| CPER | f53 | 0.04 | -0.03 | 0.02 | -0.09 | 0.09 | 0.05 | -0.08 | 1.00 | -0.16 | -0.13 | -0.03 | 0.05 | -0.04 | -0.03 | 0.06 |
| CPER | f54 | 0.05 | -0.03 | -0.06 | 0.08 | 0.17 | 0.05 | -0.11 | -0.16 | 1.00 | 0.01 | -0.05 | -0.05 | 0.01 | -0.03 | 0.10 |
| CPER | f64 | 0.01 | 0.01 | -0.07 | 0.06 | 0.00 | 0.02 | 0.03 | -0.13 | 0.01 | 1.00 | -0.02 | 0.01 | -0.03 | 0.01 | 0.00 |
| CPER | f45 | 0.02 | -0.02 | -0.05 | 0.00 | 0.18 | -0.04 | -0.06 | -0.03 | -0.05 | -0.02 | 1.00 | 0.01 | -0.11 | -0.01 | 0.07 |
| CPER | f65 | 0.02 | -0.02 | -0.04 | 0.01 | 0.05 | 0.05 | -0.02 | 0.05 | -0.05 | 0.01 | 0.01 | 1.00 | 0.02 | -0.03 | 0.01 |
| CPER | f46 | -0.02 | 0.00 | 0.00 | 0.10 | -0.03 | 0.03 | -0.02 | -0.04 | 0.01 | -0.03 | -0.11 | 0.02 | 1.00 | 0.01 | -0.01 |
| CPER | msc ut | 0.17 | 0.50 | 0.08 | -0.03 | 0.03 | 0.00 | -0.02 | -0.03 | -0.03 | 0.01 | -0.01 | -0.03 | 0.01 | 1.00 | -0.03 |
| CPER | Q10 | -0.16 | -0.47 | -0.02 | 0.01 | -0.13 | -0.02 | 0.08 | 0.06 | 0.10 | 0.00 | 0.07 | 0.01 | -0.01 | -0.03 | 1.00 |
| | | | | | | | | | | | | | | | | |
| HPG | c1 | 1.00 | -0.11 | 0.21 | 0.16 | -0.10 | -0.04 | 0.03 | -0.07 | -0.22 | 0.13 | -0.17 | -0.01 | -0.08 | -0.09 | 0.01 |
| HPG | c2 | -0.11 | 1.00 | 0.04 | -0.04 | 0.46 | -0.01 | 0.06 | 0.08 | 0.11 | -0.06 | 0.02 | -0.07 | 0.02 | 0.46 | -0.44 |
| HPG | c3 | 0.21 | 0.04 | 1.00 | -0.07 | -0.06 | -0.02 | -0.12 | -0.13 | -0.14 | 0.05 | -0.01 | 0.00 | -0.11 | 0.01 | -0.03 |
| HPG | c4 | 0.16 | -0.04 | -0.07 | 1.00 | -0.07 | -0.14 | -0.13 | 0.02 | -0.05 | -0.01 | -0.05 | 0.14 | 0.04 | 0.02 | 0.06 |
| HPG | c5 | -0.10 | 0.46 | -0.06 | -0.07 | 1.00 | 0.04 | 0.43 | 0.16 | 0.57 | 0.05 | 0.39 | -0.10 | 0.12 | 0.02 | -0.47 |
| HPG | c6 | -0.04 | -0.01 | -0.02 | -0.14 | 0.04 | 1.00 | 0.21 | 0.05 | 0.14 | 0.12 | 0.03 | -0.12 | 0.11 | 0.04 | 0.05 |
| HPG | f43 | 0.03 | 0.06 | -0.12 | -0.13 | 0.43 | 0.21 | 1.00 | 0.05 | 0.17 | -0.03 | 0.06 | -0.17 | 0.11 | 0.04 | -0.01 |
| HPG | f53 | -0.07 | 0.08 | -0.13 | 0.02 | 0.16 | 0.05 | 0.05 | 1.00 | 0.01 | -0.13 | -0.13 | -0.09 | 0.14 | -0.02 | -0.11 |
| HPG | f54 | -0.22 | 0.11 | -0.14 | -0.05 | 0.57 | 0.14 | 0.17 | 0.01 | 1.00 | 0.06 | 0.08 | -0.17 | 0.11 | 0.11 | -0.01 |
| HPG | f64 | 0.13 | -0.06 | 0.05 | -0.01 | 0.05 | 0.12 | -0.03 | -0.13 | 0.06 | 1.00 | 0.15 | 0.02 | 0.00 | -0.02 | 0.04 |
| HPG | f45 | -0.17 | 0.02 | -0.01 | -0.05 | 0.39 | 0.03 | 0.06 | -0.13 | 0.08 | 0.15 | 1.00 | -0.02 | 0.01 | -0.04 | -0.06 |
| HPG | f65 | -0.01 | -0.07 | 0.00 | 0.14 | -0.10 | -0.12 | -0.17 | -0.09 | -0.17 | 0.02 | -0.02 | 1.00 | -0.11 | 0.00 | 0.07 |
| HPG | f46 | -0.08 | 0.02 | -0.11 | 0.04 | 0.12 | 0.11 | 0.11 | 0.14 | 0.11 | 0.00 | 0.01 | -0.11 | 1.00 | 0.01 | -0.01 |
| HPG | msc ut | -0.09 | 0.46 | 0.01 | 0.02 | 0.02 | 0.04 | 0.04 | -0.02 | 0.11 | -0.02 | -0.04 | 0.00 | 0.01 | 1.00 | 0.54 |
| HPG | Q10 | 0.01 | -0.44 | -0.03 | 0.06 | -0.47 | 0.05 | -0.01 | -0.11 | -0.01 | 0.04 | -0.06 | 0.07 | -0.01 | 0.54 | 1.00 |
| | | | | | | | | | | | | | | | | |
| HAR | c1 | 1.00 | 0.86 | 0.55 | 0.04 | 0.45 | 0.10 | -0.03 | -0.10 | -0.24 | 0.00 | 0.00 | -0.06 | 0.00 | 0.66 | -0.45 |
| HAR | c2 | 0.86 | 1.00 | 0.63 | 0.03 | 0.51 | 0.08 | -0.10 | -0.10 | -0.29 | 0.00 | 0.00 | -0.07 | 0.03 | 0.76 | -0.48 |
| HAR | c3 | 0.55 | 0.63 | 1.00 | 0.02 | 0.32 | 0.01 | -0.01 | -0.06 | -0.13 | 0.00 | 0.05 | -0.08 | 0.03 | 0.48 | -0.33 |
| HAR | c4 | 0.04 | 0.03 | 0.02 | 1.00 | -0.01 | -0.06 | -0.18 | -0.02 | 0.15 | 0.05 | -0.13 | 0.11 | 0.08 | 0.03 | 0.01 |
| HAR | c5 | 0.45 | 0.51 | 0.32 | -0.01 | 1.00 | 0.03 | 0.25 | -0.10 | 0.23 | 0.04 | 0.23 | -0.04 | -0.03 | 0.31 | -0.36 |
| HAR | c6 | 0.10 | 0.08 | 0.01 | -0.06 | 0.03 | 1.00 | 0.12 | 0.01 | -0.06 | -0.06 | 0.01 | -0.05 | -0.28 | 0.02 | -0.13 |
| HAR | f43 | -0.03 | -0.10 | -0.01 | -0.18 | 0.25 | 0.12 | 1.00 | -0.30 | -0.01 | 0.09 | 0.13 | -0.10 | -0.21 | -0.12 | 0.03 |

| | | | | | | | | | | | | | | | |
|---|---|---|---|---|---|---|---|---|---|---|---|---|---|---|---|
| HAR | f53 | -0.10 | -0.10 | -0.06 | -0.02 | -0.10 | 0.01 | -0.30 | 1.00 | -0.06 | -0.12 | -0.02 | -0.01 | -0.06 | -0.03 | 0.12 |
| HAR | f54 | -0.24 | -0.29 | -0.13 | 0.15 | 0.23 | -0.06 | -0.01 | -0.06 | 1.00 | -0.05 | -0.11 | 0.03 | -0.01 | -0.16 | 0.28 |
| HAR | f64 | 0.00 | 0.00 | 0.00 | 0.05 | 0.04 | -0.06 | 0.09 | -0.12 | -0.05 | 1.00 | 0.08 | 0.17 | -0.04 | -0.02 | -0.02 |
| HAR | f45 | 0.00 | 0.00 | 0.05 | -0.13 | 0.23 | 0.01 | 0.13 | -0.02 | -0.11 | 0.08 | 1.00 | -0.12 | 0.04 | -0.01 | 0.00 |
| HAR | f65 | -0.06 | -0.07 | -0.08 | 0.11 | -0.04 | -0.05 | -0.10 | -0.01 | 0.03 | 0.17 | -0.12 | 1.00 | -0.10 | -0.06 | 0.01 |
| HAR | f46 | 0.00 | 0.03 | 0.03 | 0.08 | -0.03 | -0.28 | -0.21 | -0.06 | -0.01 | -0.04 | 0.04 | -0.10 | 1.00 | 0.03 | -0.02 |
| HAR | msc ut | 0.66 | 0.76 | 0.48 | 0.03 | 0.31 | 0.02 | -0.12 | -0.03 | -0.16 | -0.02 | -0.01 | -0.06 | 0.03 | 1.00 | 0.12 |
| HAR | Q10 | -0.45 | -0.48 | -0.33 | 0.01 | -0.36 | -0.13 | 0.03 | 0.12 | 0.28 | -0.02 | 0.00 | 0.01 | -0.02 | 0.12 | 1.00 |
| | | | | | | | | | | | | | | | | |
| KNZ | c1 | 1.00 | 0.52 | 0.04 | -0.07 | 0.28 | -0.07 | -0.03 | 0.00 | -0.07 | -0.02 | 0.02 | 0.03 | 0.01 | 0.04 | -0.48 |
| KNZ | c2 | 0.52 | 1.00 | 0.01 | -0.13 | 0.55 | -0.05 | -0.07 | -0.02 | -0.10 | -0.06 | 0.03 | 0.04 | 0.02 | 0.08 | -0.93 |
| KNZ | c3 | 0.04 | 0.01 | 1.00 | -0.05 | 0.08 | 0.09 | 0.10 | 0.04 | 0.02 | -0.12 | 0.03 | 0.01 | 0.05 | 0.04 | -0.01 |
| KNZ | c4 | -0.07 | -0.13 | -0.05 | 1.00 | -0.21 | -0.02 | -0.05 | 0.11 | 0.03 | 0.08 | -0.12 | -0.04 | -0.07 | -0.09 | 0.11 |
| KNZ | c5 | 0.28 | 0.55 | 0.08 | -0.21 | 1.00 | -0.05 | 0.19 | 0.08 | 0.23 | -0.02 | 0.34 | 0.13 | 0.03 | 0.17 | -0.48 |
| KNZ | c6 | -0.07 | -0.05 | 0.09 | -0.02 | -0.05 | 1.00 | 0.02 | 0.06 | 0.12 | -0.10 | 0.02 | -0.08 | -0.03 | -0.02 | 0.03 |
| KNZ | f43 | -0.03 | -0.07 | 0.10 | -0.05 | 0.19 | 0.02 | 1.00 | -0.02 | 0.05 | 0.10 | 0.01 | 0.01 | -0.01 | -0.04 | 0.07 |
| KNZ | f53 | 0.00 | -0.02 | 0.04 | 0.11 | 0.08 | 0.06 | -0.02 | 1.00 | 0.05 | 0.09 | 0.05 | -0.13 | -0.12 | 0.04 | 0.04 |
| KNZ | f54 | -0.07 | -0.10 | 0.02 | 0.03 | 0.23 | 0.12 | 0.05 | 0.05 | 1.00 | 0.00 | -0.12 | -0.02 | -0.05 | -0.02 | 0.09 |
| KNZ | f64 | -0.02 | -0.06 | -0.12 | 0.08 | -0.02 | -0.10 | 0.10 | 0.09 | 0.00 | 1.00 | -0.14 | 0.07 | -0.11 | 0.05 | 0.06 |
| KNZ | f45 | 0.02 | 0.03 | 0.03 | -0.12 | 0.34 | 0.02 | 0.01 | 0.05 | -0.12 | -0.14 | 1.00 | 0.08 | 0.14 | 0.09 | -0.02 |
| KNZ | f65 | 0.03 | 0.04 | 0.01 | -0.04 | 0.13 | -0.08 | 0.01 | -0.13 | -0.02 | 0.07 | 0.08 | 1.00 | 0.00 | -0.07 | -0.06 |
| KNZ | f46 | 0.01 | 0.02 | 0.05 | -0.07 | 0.03 | -0.03 | -0.01 | -0.12 | -0.05 | -0.11 | 0.14 | 0.00 | 1.00 | 0.01 | -0.02 |
| KNZ | msc ut | 0.04 | 0.08 | 0.04 | -0.09 | 0.17 | -0.02 | -0.04 | 0.04 | -0.02 | 0.05 | 0.09 | -0.07 | 0.01 | 1.00 | 0.16 |
| KNZ | Q10 | -0.48 | -0.93 | -0.01 | 0.11 | -0.48 | 0.03 | 0.07 | 0.04 | 0.09 | 0.06 | -0.02 | -0.06 | -0.02 | 0.16 | 1.00 |




**Table B4. Descriptions of soil profile data sets obtained from the International Soil Carbon Network**

| Location | Site code | Latitude | Longitude | Profile depth (cm) |
|---|---|---|---|---|
| Sevilleta National Wildlife Refuge, New Mexico | 81NM053005 | 34.18703 | -107.21 | 152 |
| Central Plains Experimental Range, Colorado | S1991CO123007 | 40.82806 | -104.786 | 130 |
| Cheyenne, Wyoming | S1989WY021007 | 41.20889 | -104.931 | 160 |
| Hays, Kansas | S1968KS051001 | 39.01528 | -99.1258 | 145 |
| Konza Priarie Biological Station, Kansas | KZP 1D | 39.07597 | -96.5638 | 200 |





**Table B5. Variance partitioning results comparing among site variation in net primary productivity explained by site-level estimates of average annual precipitation, average annual temperature, soil bulk density, grass:forb, C3:C4, and annual species abundance.**

| Variable 1 | | Variable2 | Test | AdjR2 | Fval | Pval |
|---|---|---|---|---|---|---|
| | | | X1\|X2 | 1.17 | 123.71 | 0.01 |
| | | MAP | X1X2 colin. | -0.21 | NA | NA |
| | | | X2\|X1 | 0.00 | 0.58 | 0.51 |
| | | | X1+X2 | 0.96 | 63.84 | <0.01 |
| | | | X1\|X2 | 0.36 | 32.70 | 0.01 |
| | | Bulk Density | X1X2 colin. | 0.61 | NA | NA |
| | | | X2\|X1 | -0.01 | 0.03 | 0.84 |
| | | | X1+X2 | 0.95 | 53.74 | 0.02 |
| | | | X1\|X2 | 0.09 | 11.44 | 0.05 |
| NPP | MAP | Grass:forb | X1X2 colin. | 0.87 | NA | NA |
| | | | X2\|X1 | 0.00 | 0.82 | 0.43 |
| | | | X1+X2 | 0.96 | 68.27 | <0.01 |
| | | | X1\|X2 | 1.15 | 103.19 | <0.01 |
| | | C3:C4 | X1X2 colin. | -0.18 | NA | NA |
| | | | X2\|X1 | -0.01 | 0.04 | 0.83 |
| | | | X1+X2 | 0.95 | 54.02 | 0.01 |
| | | | X1\|X2 | 0.50 | 74.70 | 0.01 |
| | | Annual species abundance | X1X2 colin. | 0.47 | NA | NA |
| | | | X2\|X1 | 0.01 | 2.06 | 0.24 |
| | | | X1+X2 | 0.97 | 90.77 | <0.01 |
| | | | X1\|X2 | 0.54 | 4.10 | 0.18 |
| | | MAP | X1X2 colin. | -0.16 | NA | NA |
| | | | X2\|X1 | -0.09 | 0.10 | 0.71 |
| | | | X1+X2 | 0.70 | 2.05 | 0.33 |
| | | | X1\|X2 | 0.22 | 2.21 | 0.22 |
| | | Bulk Density | X1X2 colin. | -0.18 | NA | NA |
| | | | X2\|X1 | 0.24 | 0.02 | 0.86 |
| | | | X1+X2 | 0.72 | 1.97 | 0.37 |
| Eco$_{RT}$ | MAT | | X1\|X2 | 0.60 | 4.74 | 0.16 |
| | | Grass:forb | X1X2 colin. | -0.10 | NA | NA |
| | | | X2\|X1 | -0.15 | 0.38 | 0.52 |
| | | | X1+X2 | 0.64 | 2.38 | 0.24 |
| | | C3:C4 | X1\|X2 | 0.22 | 2.22 | 0.23 |
| | | | X1X2 colin. | -0.18 | NA | NA |

|  |  |  |  |  |
|---|---|---|---|---|
|  | X2\|X1 | 0.23 | 0.01 | 0.95 |
|  | X1+X2 | 0.73 | 1.95 | 0.25 |
| Annual species abundance | X1\|X2 | 0.36 | 3.16 | 0.22 |
|  | X1X2 colin. | -0.12 | NA | NA |
|  | X2\|X1 | 0.10 | 0.29 | 0.69 |
|  | X1+X2 | 0.66 | 2.27 | 0.30 |





**Table B6. Average soil moisture and soil temperature during June-September at our six focal grassland sites.**

| Site | Volumetric soil moisture (%) | Soil temperature (°C) |
|------|------|------|
| SBL | 13.3 | 26.3 |
| SBK | 11.9 | 27.6 |
| CPER | 11.4 | 22.9 |
| HPG | 15.4 | 21.1 |
| HAR | 21.0 | 24.6 |
| KNZ | 20.9 | 24.0 |





## Appendix C. Detailed description of data collection and cleaning methods

### C1 Aboveground net primary productivity (ANPP)

At CPER, HPG, HAR, and KNZ, ANPP was collected annually by clipping two 0.1 m$^2$ subplots per replicate in September, sorted to remove previous years' growth, dried at 60 C for 48 hours, and weighed. At SBK and SBL, ANPP was estimated using species-specific allometric methods (Muldavin et al. 2008, Rudgers et al. 2019) within four 1 m$^2$ subplots within each of the 10 control plots.

### C2 Vegetative litter

Litter estimates were obtained by collecting previous year's biomass to ground level in the same subplots as ANPP. Litter was not present at SBL or SBK due to rapid decomposition at the soil surface at these sites, nor at KNZ due to annual burning.

### C3 Belowground net primary productivity (BNPP) and root standing crop

BNPP was estimated from 0-20 cm at each site using two root ingrowth cores per plot (Persson 1980). Ingrowth cores were constructed from 2 mm fiberglass screen molded into a 5 cm diameter 22 cm long cylinder (2 cm was left above soil surface). In April-May, a soil auger was used to drill 20 cm deep in the soil. Ingrowth cores were then inserted, filled with sieved, root free soil from the site, and packed to approximate soil density. Ingrowth cores were removed in late September and stored at 4 C until processing. Samples were elutriated to separate root biomass from soil. Samples were sorted to remove soil organic matter, dried at 60 C for 48 hours, and weighed. Finally, samples were fired at 450 C and resulting ash subtracted from the biomass value to generate an ash free estimate of BNPP. 5 cm root standing crop samples were also taken from 0-20 cm in two subplots. Standing crop root samples were processed similarly to BNPP samples, with the exception that dead roots were sorted out of the samples at the same time as SOM.

### C4 Soil CO$_2$ efflux

Surface soil respiration was calculated using measurements of soil CO$_2$ concentrations (GMP 220 series probes, Vaisala Corp., Helsinki, Finland) in three replicates per site at 5, 10, and 20 cm depths, combined with diffusion rates calculated using soil temperature and moisture data collected simultaneously with CO$_2$ concentration. See Vargas & Allen (2008), and Vargas et al. (2010) for more information about calculating soil respiration rates using this method. We averaged 15 minute soil respiration measurements between 10:00 am and 2:00 pm to obtain daily soil respiration values.

**C5 Soil moisture and temperature**

Volumetric soil moisture integrated from 0-15 cm was measured in all 10 plots every 15 minutes using time domain reflectometry probes (CS-616 model, Campbell Scientific, Inc., Logan, UT, USA). Soil temperature was measured in 3 plots at 5 cm and 10 cm depths using thermocouples (K-type, OMEGA Engineering Inc., Stamford, CT, USA).

**C6 Soil bulk density**

Bulk density was measured at each site from ten 7 cm diameter soil cores taken 0-30 cm. Soil cores were extracted in segments to reduce compaction. All aboveground vegetation was removed, and cores were dried at 105 C for 48 hours, then weighed. Bulk density was then calculated as:

$$BD = \frac{M_{dry}}{\pi r^2 x\, D}$$ (C1)

Where $M_{dry}$ is the dry mass, $r$ is the radius, and $D$ is the depth of the soil core.

**C7 Meteorological data**

Hourly air temperature, precipitation, relative humidity, vapor pressure deficit (VPD), and incident photosynthetically active radiation data were obtained from nearby weather stations to run the terrestrial ecosystem model (TECO; Weng and Luo, 2008). Gaps of $< 10$ time steps (hours) were filled by splining, and all splined sections examined to ensure values were within reasonable bounds. Gaps of longer time steps were filled from next closest meteorological stations of the same 820 elevation. When unavailable, we calculated VPD as actual vapor pressure ($e_a$) minus saturation vapor pressure ($e_s$). $e_a$ is estimated as:

$$e_a = \frac{RH}{100} e_s$$ (C2)

where RH is relative humidity, and $e_s$ is estimated as:

$$e_s = 0.6108 e^{\frac{17.27t}{t+237.3}}$$ (C3)

where $t$ is air temperature.

**C8 Soil total organic carbon**

At CPER, HGR, HAR, and KNZ, percent C was measured in three 2 cm diameter 10 cm deep soil cores per plot. Subsamples were aggregated, sieved with a 2 mm soil sieve to remove root material, dried, and ground. Samples were then

measured for total C via dry combustion and grass chromatography using a LECO CN 2000 combustion analyzer (LECO Corp., Saint Joseph, MI, USA). For SBL and SBK, percent organic matter was estimated in soil from 0-10 cm depths from nearby areas having similar soils and vegetation to the EDGE sites. Organic matter values were then converted to percent C by dividing the percent organic matter by 1.72 (assumes 58% C stoichiometry of organic matter). Percent C for all sites was then converted to total C using:

$$C_T = D \ x \ BD \ x \ C_p x \ 10000 \tag{C4}$$

Where $C_T$ is total carbon, $D$ is the depth in cm to which the sample was collected, $BD$ is the bulk density in g per cm$^{-2}$, and $C_p$ is the proportion of carbon.

**C9 Plant species abundance**

Abundance of plant species was estimated visually in June and August in four 1 x 1 m$^2$ subplots per plot to the nearest percent. Species having less than 1% cover were rounded up to 1%. Calibrations of visual estimates were performed across researchers and with measurements seasonally. Maximum cover of each species was taken for each subplot, then all abundances were averaged across subplots to get plot-level estimates.

**Appendix D. Model evaluation text, Table D1, Figures D1-D3**

Model validation exercises were conducted to compare the ability of the model to represent (1) average plant growth across sites and (2) variability of plant growth across years within each site. To this end, models were parameterized for each site based on soil texture, field capacity, wilting point, latitude, and root:shoot ratios. Additionally, maximum and minimum specific leaf area and Vcmax parameters were adjusted to better match measured above and belowground plant growth. Table 1 shows the parameter sets used for each site.

Empirical observations of ANPP and BNPP (n=10) were compared to leaf+stem biomass and root productivity from the model at the time of data collection. At CPER, HPG, HAR, and KNZ, ANPP was collected annually by clipping two 0.1 m2 subplots per replicate in September, sorted to remove previous years' growth (leaf litter), dried at 60 C for 48 hours, and weighed. At SBK and SBL, ANPP was estimated using species-specific allometric methods (Muldavin et al. 2008) within four 1m2 subplots within each of the 10 replicates. BNPP within each replicate at each site was estimated from 0-20 cm using two 20 cm x 5 cm diameter root ingrowth cores per plot (Persson 1980). See Appendix 3 for additional details about empirical data collection.

Models were spun up for 500 years using cycled meteorological data from each site – all carbon pools stabilized after 200-400 years. Then simulations were conducted for 2014-2017 for all sites. Empirical ANPP data were available from 2014-2017 for all sites, and BNPP was available for CPER, HPG, HAR, and KNZ for 2014-2017. BNPP was available at SBL and SBK from 2015-2017. To assess model performance, we calculated various validation metrics, including $R^2$, RMSE, and the correlation coefficient.

Generally, cross-site averages of model output and observations were tightly correlated, but within-site model output was less correlated to interannual empirical observations (Table 1). However, it is important to note the variation associated with empirical measurements and that most model simulations fell within one standard error of empirical observations (error bars in Fig. D2 and D3). A notable exception to this occurred in 2014 at CPER and HPG sites, where BNPP was observed as much greater than model simulations. We are unsure what drove the high BNPP at these sites in 2014 since environmental conditions were within normal ranges (Fig. D1). It could be that other unmeasured variables (e.g., belowground animal activity, small mammal activity) were responsible for this high growth belowground.

**Table D1. Model output for comparisons between empirical and simulated data.**

| Site | Productivity type | R2 | RMSE | Correlation coefficient |
|------|------|------|------|------|
| means | anpp | 0.988738 | 17.96882 | 0.994353 |
| means | bnpp | 0.93693 | 23.4836 | 0.967952 |
| sbl | anpp | 0.431519 | 9.568253 | -0.6569 |
| sbl | bnpp | 0.009114 | 18.25828 | -0.09547 |
| sbk | anpp | 0.667188 | 10.51358 | 0.816816 |
| sbk | bnpp | 0.059729 | 13.3254 | -0.2444 |
| cper | anpp | 0.087017 | 33.30121 | 0.294986 |
| cper | bnpp | 0.181185 | 98.48398 | 0.425659 |
| hpg | anpp | 0.018955 | 35.10083 | -0.13768 |
| hpg | bnpp | 0.022188 | 131.1703 | 0.148957 |

| | | | | |
|---|---|---|---|---|
| har | anpp | 0.526963 | 28.82132 | 0.725922 |
| har | bnpp | 0.330895 | 34.27893 | 0.575235 |
| knz | anpp | 0.004173 | 162.8426 | -0.0646 |
| knz | bnpp | 0.637154 | 36.67331 | -0.79822 |

anpp=aboveground net primary productivity, bnpp=belowground net primary productivity, sbl=Sevilleta National Wildlife Refuge blue grama grassland; sbk=Sevilleta National Wildlife Refuge black grama grassland; cper=Central Plains Experimental Range; hpg=High Plains Grasslands Research Station; har=Hays Agricultural Research Center; knz=Konza Prairie Biological Station

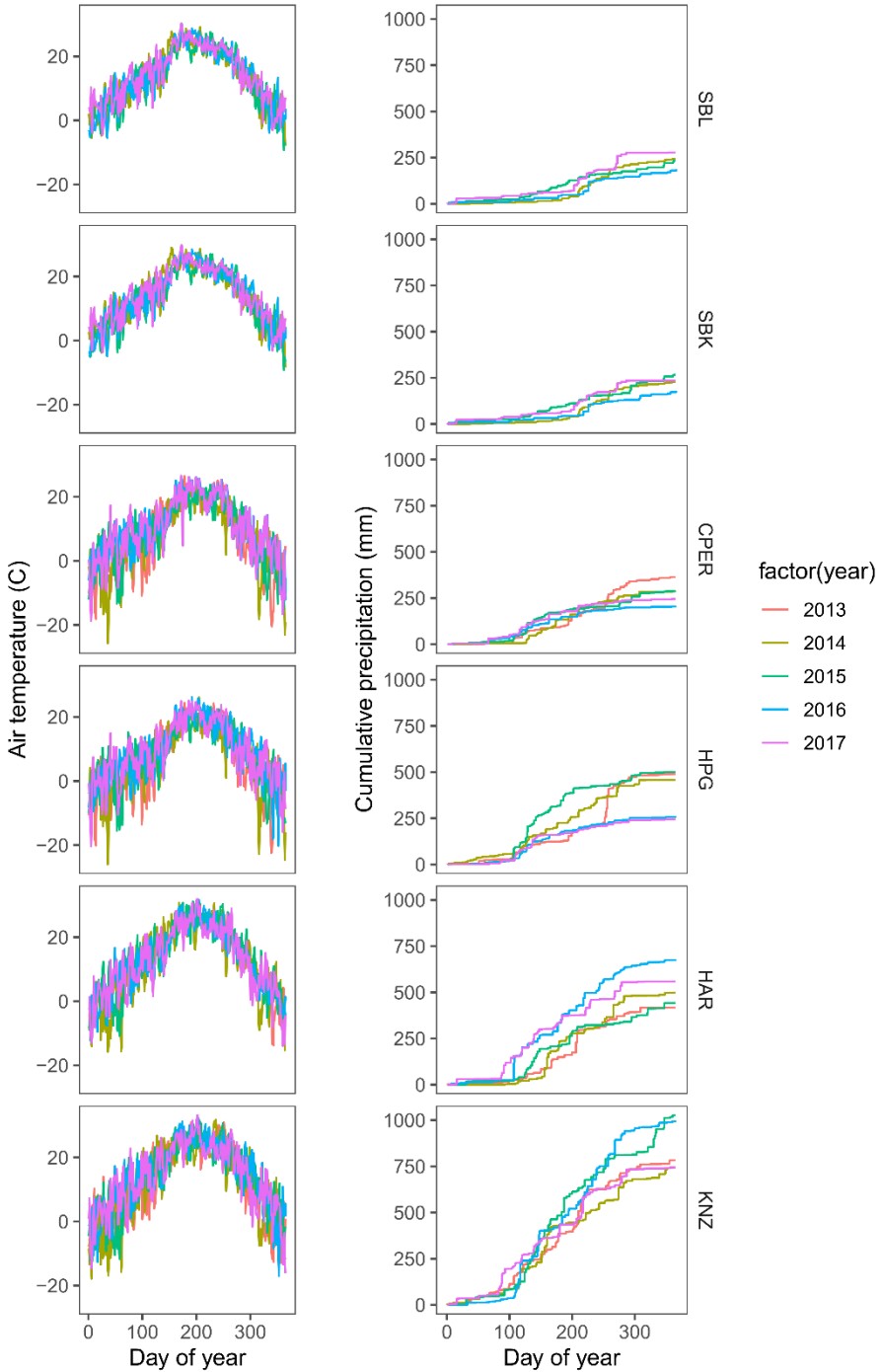

**Figure D1. Cumulative hourly precipitation and hourly air temperature throughout each year simulated within each site.**


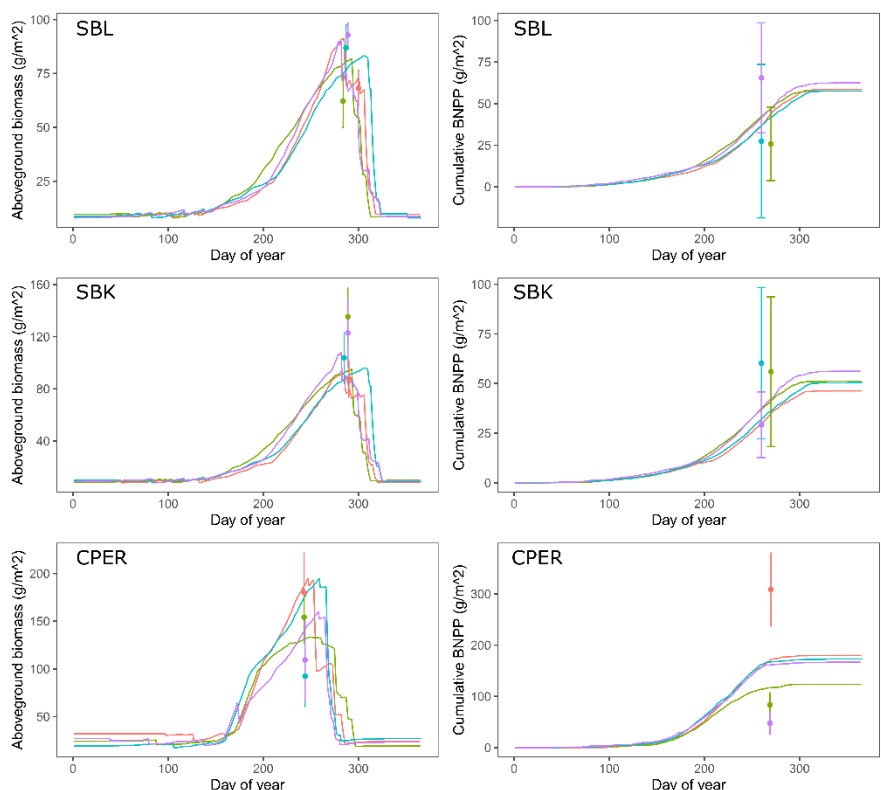

**Figure D2. Within year patterns of simulated aboveground biomass and BNPP for SBL, SBK, and CPER, compared with empirical observations. Bars represent one standard deviation from the mean of the 10 replicate measurements. Years are represented as colors using the scale from Fig. 1.**



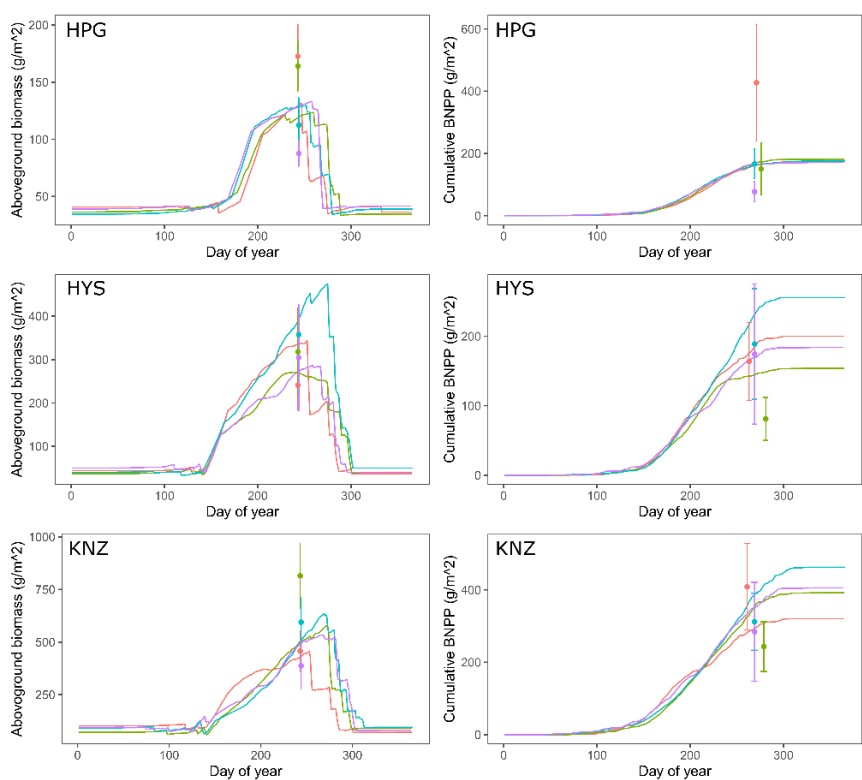


**Figure D3. Within year patterns of simulated aboveground biomass and BNPP for HPG, HYS, and KNZ, compared with empirical observations. Bars represent one standard deviation from the mean of the 10 replicate measurements. Years are represented as colors using the scale from Fig. 1.**



## Appendix E. Land use history of study sites

Here, we outline the recent land use history starting with the establishment of the current associations responsible for managing these areas.

SBL, SBK: The Sevilleta National Wildlife Refuge in central New Mexico, USA, was established in 1973. Prior to that time the land was lightly to moderately grazed by domestic herbivores (mostly cattle along with some horses and sheep) for about three centuries as part of a collective land grant to settlers from the King of Spain. All domestic herbivores were removed
from the site when the refuge was established, and the vegetation has recovered from grazing (Gosz and Gosz 1996, Collins and Xia 2015). The refuge sits at the intersection of several biomes (Gosz and Gosz 1996) including Great Plains grassland dominated by blue grama (Bouteloua gracilis)(SBL site in this study) and Chihuahuan Desert grassland dominated by black grama (B. eriopoda)(SBK site in this study). Chihuahuan Desert grassland is advancing northward into Great Plains grassland (Collins et al. 2020) in response to increasing aridity throughout the southwestern US (Rudgers et al. 2018, Maurer
et al. 2020).

CPER: The Central Plains Experimental Range is a shortgrass prairie dominated by B. gracilis (Hazlett 1998). The experimental station was established in the 1930s to better understand how to manage lands to avoid catastrophic occurrences such as the Dust Bowl. A number of climatic and management effects occurred at CPER, including a severe blizzard in 1949, an extreme flood in 1965, and sever droughts in 1939, 1954, 1964 (Shoop et al. 1989), and 2012 (Knapp et
al. 2015). The specific area where our plots were located have been ungrazed for 14 years prior to the start of measurements. Before then, grazing occurred in the area but fire was largely absent at the site.

HPG: The High Plains Grassland Research Station was authorized by congress in 1928 to experiment with various horticultural crops in arid lands, which was begun in 1930. In 1974, the station shifted focus to rangeland management, water conservation, and land reclamation. The area where the plots were located was ungrazed since 2004 (Dijkstra et al. 2010)
and has little to no history of fire.

HAR: Kansas State University began management of The Agricultural Research Center-Hays in 1994 and was managed for cattle grazing. The plots where our plots were located were ungrazed and unburned since 2005 (Heisler-White et al. 2009).

KNZ: The land where Konza Prairie Biological Station resides was purchased by the Nature Conservancy in 1971 and deeded to Kansas State University to establish a watershed-scale experiment to assess the impacts of fire and grazing on
tallgrass ecological processes (Knapp et al. 1998). The watershed where our measurements took place has been burned annually in the spring since 1988 and was ungrazed by large ungulates since the early 1980s.

## Code availability

R scripts to conduct data assimilation, calculate all metrics presented in the paper and statistical analyses publically available at https://github.com/wilcoxkr/AssessingCarbonCapacity. TECO model code is available on the ECOLAB website here: https://www2.nau.edu/luo-lab/?downloads.

## Data availability

MCMC chains and raw empirical data is available upon request to the corresponding author.


**Author contributions**

KRW, YL, MDS, SC, WP, and AKK designed the research project and participated in idea generation; KRW and ZS conducted analyses and model simulations; KRW led the writing process; all authors contributed to writing and editing process.

**Competing interests**

The authors declare that they have no conflict of interest

**Acknowledgements**

Support was provided to MDS and AKK by the Drought-Net Research Coordination Network funded by the US National Science Foundation (DEB-1354732), the Long Term Ecological Research Program of the US National Science Foundation (DEB-1655499), and by the Macrosystems Biology/Emerging Frontiers Programs (EF-1239559, EF-1137378). Support was provided to KRW by the Department of Energy (DE-SC0019037). Support was provided to SLC by NSF award DEB-1856383.

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
