# Peer review of "Assessing carbon storage capacity and saturation across six central US grasslands using data-model integration"

_Biogeosciences, 2022_

## Author Response (AR1)

**Reviewer 1**:

This is an interesting paper, examining the potential carbon storage capacity of grasslands.  Overall, it is well-written, and provides interesting and important results. That said, there are many limitations.

*Thank you. We feel that the revisions undergone since the first version of the manuscript have resulted in a substantially improved manuscript.*

First, the number of abbreviations, acronyms, and symbols make it almost unreadable.  I understand that the authors are working to present efficiently, but it's too much.  The paper would benefit from a major re-writing, using real words instead of writing sentences and even research questions and interpretations about CSAT, Xp, XC, TE, etc.  Most of us know NPP, MAT, and MAT, but still, these others in long complicated sentences make the paper impossible to really understand without having the glossary at hand.

*In the manuscript, we tried to balance efficiency with readability, yet we recognize that it made the text difficult to follow. We have replaced most of the abbreviations with descriptive text and we now include a new table (Table 1) describing each term and how it is derived. We think that this not only improved the flow of the text, but it also helps with other concerns raised about difficulties understanding how each of these terms were estimates (e.g., model vs empirical).*

The list of "questions and associated hypotheses" is sloppy.  The first one has been tested many times over (and they are very short on citations about this); the second is not even close to a hypothesis (they should either use questions or hypotheses instead of garbling them up together); the third doesn't have a causal explanation, necessary for a hypothesis (a prediction with a causal explanation). As I understand it Csat is the proportion of current carbon storage relative to the potential carbon storage, and the explanation doesn't relate to this. That third one is an example of an idea that a) doesn't make sense, and b) is impossible to interpret given the density of acronyms/symbols.

*This is another instance where brevity was our goal. Instead of having separate sections for questions and hypotheses, we decided to combine them. This sometimes resulted in questions without associated hypotheses (e.g., Q2) when we felt the current understanding was not sufficient to posit meaningful hypotheses. We agree that Q1 about relationships of NPP, $\tau E$, and $XP$ with MAP and MAT has been the focused on in previous studies. Yet, we argue that there is value to having both questions that test current ideas along with questions that test entirely new ideas, thus grounding novel work with more established patterns. And also, we find that some of the commonly found patterns do not show up here, and we feel that is important. However, we have reworked the text in this section so that there are now questions and associated predictions for each of the three major lines of inquiry in the study. Along with this, we completely revised Qs 2 and 3.*

It's quite difficult to tell from the Methods which of the values were modeled and which were using the measured data from the sites.

*We apologize that the Methods were unclear about the sources of the values used in the paper. Wherever possible, we used empirical data to inform our analyses. However, much of the time, empirical data did not capture the fullness of information necessary for calculating holistic variables, such as carbon potential. We have revised the text to be more explicit about where each piece of information came from. This is done throughout the methods and in our new Table 1.*

The explanation of "formal model validation" states that they validated the vegetation components. Notably, vegetation does not comprise the major pools of C in these systems.

*We agree. This, along with difficulties present in measuring C cycling parameters in the field (e.g. slow C turnover), is why we used the data assimilation approach to estimate the processes that drive soil C pools. We used high temporal resolution C pool and flux data to estimate 15 parameters associated with the C submodel (Fig. A1-4, Table B1), which we feel is in itself a valuable contribution to the field. We argue that this approach is a much more robust method of estimating C cycling pools and processes than simple model calibration and validation, since there could be many ways to achieve the observed carbon flux values (e.g. fast inherent turnover times, high temperature sensitivity), and not all of these may be correct. Although not failsafe, the data assimilation approach is less apt to arrive at the right answer (i.e. close to benchmark data) for the wrong reasons (i.e. by adjusting the wrong parameters) – this is because all of these parameters are allowed to vary independently during the data assimilation process. We have greatly expanded the text associated with this process in the methods.*

There are some problems (line 195) with this method of estimating carbon storage with depth, as this varies so much by soil profile and location - it's not terrible, but a caveat should express the limitaitons of the approach.

*The reviewer makes a good point about difficulties estimating carbon storage by depth. However, we do think it is important to estimate soil C from 0-20 cm to ensure that simulated and observed C pools match in the layers of soil they are estimated in. We now include text about these uncertainties at the end of the discussion.*

I don't understand the "normalization" nor which slopes the authors are referring to (line 210).

*We mean that the values were scaled by the standard deviation and centered around the mean, we now better explain this in the text.*

For a number of these systems, a large proportion of the carbon stored is in recalcitrant soil pools. IThere needs to be MUCH more citation and analysis - and probably reconsideration of these residence times. Previous authors have shown that a good portion of the ecosystem carbon in these grasslands turns over on thousand year time scales, not time scales of 20 or so to 50 years. This alone gives me a great deal of concern about the paper. The paper should at least note that their estimates are orders of magnitude less than others have published.

*We appreciate this comment and have incorporated discussion of previous literature that address these ideas, and we also have expanded the discussion about the effects of changes in these slower pools ('slow' and 'passive' pools in TECO) on carbon residence times from our analyses. We would like to note that these carbon residence time estimates incorporate all carbon substrates including both those that turn over quickly and those that turn over much more slowly. If we only focus on the recalcitrant substrates from our study here, we see inherent turnover rates more in line (167-194 years) with the comment here by the reviewer, albeit not quite thousands of years. The discrepancy may be due to our focus on relatively shallow soil layers. In addition to expading the discussion of the effects of these pools in the discussion section of the MS, we also have included some caveats in focusing on the top 20 cm of the soil.*

*To address the uncertainty associated with our estimates of C saturation, we conducted a new bootstrapping analysis where we incorporated uncertainty in C residence time, NPP, and present C into our estimates of C saturation. This is shown in the updated Figure 5. We think this has been a great addition to the manuscript because it shows the incorporation of uncertainty all the way from the data-informed estimates of model parameters to the final calculations of C saturation.*

The paper misses alot of literature about carbon storage, NPP, and decomposition across the region - it is almost shocking.  There are very solid papers on the trends in soil carbon storage of grasslands vs. croplands (disturbed systems) across the central grasslands region,  and on the trends in NPP and decomposition (k values) tested against mean annual precipitation and mean annual temperature that are never cited, in addition to other papers addressing mechanisms of C storage across the grasslands gradients in the region, in large scale databases and in a very original and key modeling paper for the region - this latter paper seems like a seriously important progenitor and the gap in citing it is pretty egregious.

*Originally, we had not included text (or associated references) for comparisons of croplands versus non-tilled grasslands since soil processes differ substantially in tilled versus untilled soil and by crop type. However, we recognize the value in this since much of the historical extent of grasslands is now cropland. We now include many of these papers in the introduction and discussion sections of the manuscript.*

*Although we do include articles associate with the mechanisms and gradients of decomposition rates (Brandt et al 2010, Garcia-Palacios et al 2016), and NPP (Sala et al. 2012, Huxman et al. ,2004), we have expanded these citations to include additional studies such as Zhou et al 2009 Ecosystems, Bontti et al 2009 GCB, Yahdjian et al 2006, Burke et al 1997 Ecology, Parton et al 1993 GBC, and others.*

*With regards to addressing previous literature stemming from modeling results and large scale databases, we assume that the reviewer is referring to the LIDET data set and associated papers (e.g., Bonan et al 2012 GCB, Adair et al 2008 GCB, Bontti et al 2009 GCB, Harmon et al 2009 GCB, Parton et al 2007 Science). This was an oversight on our part – we appreciate the reviewer pointing this out and we now address this previous work in both the introduction and discussion.*

The paper does not really address the key issues about the effects of most disturbances (ie the distance between Xp and Xc, ack) on soil carbon storage, or what really happens that reduces C storage. It's quite theoretical, which is ok, but ungrounded in other literature on carbon storage either from empirical work or from modeling work in these grasslands.

*In the simulation of soil carbon capacity, the effect of disturbance is primarily through the loss of NPP inputs due to losses during disturbance. We have now included citations for Ojima et al Biogeo 1994, Lorenz and Lal C Seq in Ag Eco 2018.*

There's no citation of where the effects of temperature on decomposition came from, from the empirical literature, used in the modeling (fig. A4 showing the modeled relationship between temperature and decomposition ) - it's as though the authors made up these relationships and values out of their heads, instead of from empirical data or others' work.

*The temperature effect on decomposition (Q10) is shown as a density distibution obtained from the data assimilation process we conducted in our study. The bounds for the parameter were obtained from density distributions presented in Shi et al. (2015 Ecosphere), but the distributions shown in Fig A4 are obtained from the Markov chain Monte Carlo simulations comparing model output with observations. We hope that our expanded description of the data assimilation process alleviates this concern.*

The conclusions are broad, and don't really present new insights.  One reason that the hot and dry grasslands may have more C than they think is "their capacity" (the gap between the Xp and Xc) ould be that the model is not representing the systems well - that should be clearly state - it may not actually be a reflection of system dynamics. Finally, the conclusion spends a lot of time on burning, a relatively rare disturbance in some of the systems studied, and consideration of other disturbances should be included.

*We respectfully disagree with the reviewer on this point. We think that this study presents major conclusions that can guide future management of grassland systems. Although the model-data fusion approach makes assumptions that may not perfectly mirror the carbon cycle in all ecosystems, it allows for estimating ecosystem attributes that are difficult or impossible to estimate without very long and detailed observational records of carbon pools and fluxes – namely, it allows us to estimate future tendancies of ecosystem to gain or lose carbon. First, the finding here that fast carbon turnover rates in drylands without increased NPP inputs may cause future C loss is important for guiding land management and future research priorities. These findings are corroborated by recent empirical findings in drylands showing increased C losses in arid grasslands linked with wetter and productive years likely through increased microbial activity (Hou et al 2021 Biogeosciences). The effect of fire for carbon potential is another important finding from our study. In many ecosystems fire enhances NPP and is used as a common management tool. There is uncertainty surrounding the balance between increases in NPP and decreases in conversion of ANPP to soil C – our study provides evidence that the losses of ANPP may eventually outweigh the increases in total NPP leading to C loss. We have expanded discussion of these ideas in the manuscript.*

This is an interesting study and a generally well-written manuscript. The authors provide a quantitative assessment of the C capacity and saturation of 6 US grasslands using both measured and modelled data.

*Thank you for the kind words and the constructive feedback. We feel that the revised version here is greatly improved, largely due to both of the reviewers' suggestions.*

The main weak points of the manuscript are (1) the inconsistent and confusing use of terms (2) the limited discussion around the role of soil biogeochemical processes for the C balance/capacity/saturation and (3) the minimal explanation of the *data assimilation process* i.e. what are the assimilated data about ?

*Here, we respond to each of the three points above:*

*(1) We agree with the reviewer and we have rewriten the text to limit the number of acronyms used and included a new table (Table 1). We think this has substantially improve clarity of the manuscript.*

*(2) This is another good point raised by the reviewer. We now include many additional references in both the introduction and discussion concerning processes that may be driving our findings here.*

*(3) Originally, we had minimized the text describing the data assimilation process, but based on this comment and some confusion surrounding Fig. 2-4, we have expanded the text in the methods and included Table 1 to better describe the data assimilation process. Briefly though, we used NPP inputs, soil temperature and soil moisture to drive the carbon turnover submodel. Then, we compared model output to observations of ANPP, root standing crop, plant litter, soil carbon, and surface $CO_2$ efflux measurements through time to optimize C turnover, transfer, and sensitivity parameters. These parameters were then used to calculate carbon capacity.*

After reading the paper carefully I am not able to explain how many key variables were estimated e.g. potential C. I can understand the key findings due to the very nice graphics. I think this reflects what the manuscript is lacking. All the elements of a good publication are in there but not given to the reader in a clear and coherent manner.

*This is a good point. We think that our revisions have clarified how the various metrics were estimated.*

**Specific comments**

- abstract : C can be lost via leaching also

  *Done*

- abstract : "The proportion of ð• ' ‹ c currently stored by an ecosystem (i.e., its C saturation – CSAT)" -- this is assuming a grassland ecosystem *is C saturated*, which is almost never the case (can be close to but not *at* Csat)

  *We agree and have updated the text (L62-66) to make this point more clear.*

- Page 2 : C capacity *Xc* and C content *Xp* become a source of confusion as there are points in the MS where *Xc* is presented as present/current C content and *Xp* as the potential C (e.g. L77)

  *We think that removing the symbols and using consistent terminology has alleviated this issue.*

- L83: *Csat* is presented a "the distance between Xp and Xc" but later referred to as "proportion" and "percentage" which leads to different readers understanding this variable very differently

  *Same as previous response – hopefully solved this issue.*

- L101 : What is the land use history (at least the recent one) of the examined sites? Where they always grasslands?

  *This is an important point. Many of the sites were abandoned grazing land so there is a legacy of that management that may still be present. We have now included how long each site has been ungrazed in the methods (L113-117) and in a new appendix which describes the land use history of each site (Appendix E).*

- Section 2.5 : I believe that all terms used in the MS should be described in one unique section early on. A table and/or schematic would help a lot.

  *We think this is a great idea, and have implemented this in Table 1*

---

## Author Response (AR2)

Reviewer #1

The authors' revisions to the manuscript (MS) in response to previous comments are adequate. My recommendation is for the MS to be accepted after minor revisions are completed. These revisions should cover two points :

1. The authors could add more text/discussion (page 19) on the role of C3 plants on dead tissue production and decomposition as those sites with C3 % > 0 clearly have different C capacity (and estimated uncertainty). Are the authors arguing that plant species composition's role on C storage capacity and saturation is "negligible"?

*We appreciate this comment and have included a new paragraph focused on the potential impacts of C3 abundance on C cycling dynamics (L 423-435). In brief, we did find that the two sites with the highest C3 abundance had faster turnover rates of aboveground plant tissue and 'fast' SOM pools. However, this didn't result in particularly low C residence times due to slower root turnover rates at one site (HPG) and low sensitivity of C turnover to temperature and soil moisture (CPER).*

2. I understand that soil $CO_2$ flux data (soil respiration) were collected and assimilated but (a) the references to the measurement method are missing and (b) the limitations of the method and the role of the uncertainty of the collected data are not discussed. Soil respiration $CO_2$ fluxes are very variable (temporally and spatially) but they are, also, a large flux of the C cycle at the ecosystem scale. This means that the uncertainty of these measured data and how it is treated in the assimilation process are too important to not discuss. For example, it would be good to see time series of measured soil respiration (with error bars) per site as well as some text on how the uncertainty around soil respiration $CO_2$ (and other assimilated variable; in fact) is integrated into the MCMC process i.e. what metric of model forecasting skill is used in the assimilation process and how does it allow for measured data uncertainty to be considered?

*We created a new supplemental figure (Fig. A9) with all the raw surface $CO_2$ flux data used to drive the data assimilation process, along with the average uncertainty associated with cross-plot estimates. We also included text in the discussion concerning this uncertainty (L506-510).*

Overall, I believe the authors have made a good job with revisions. With some additional minor revisions the MS would be more complete. Finally, I believe that the authors could refer to the relevant literature more when building their case for data-model integration. It is not necessary to do so but it can make the article a better read. The number and quality of studies on ecosystem C cycling quantification through data assimilation has been increasing since ~2010. Fusing model predictions with data measured at ground level and obtained via UAV/airplane/satellite-born instruments improves the robustness and the spatial/temporal resolution of C cycling estimates.

*We appreciate this suggestion and now have expanded our discussion of data-model fusion into the introduction L85-90.*

Reviewer #2

The manuscript is much improved after its response to reviewer comments.

*We appreciate the kind words.*

There is much greater clarity thanks to less reliance on study specific variables and the addition of a new Table 1. However, the same issue remains when referring to the model parameters. I suggest that giving a description on first use in each major section, i.e. don't rely on the reader having read all of the methods to find the definition of mscut, as the paper is still quite dense. Similar issues can be found in the discussion.

*We incorporated this suggestion two ways. First, we now have updated the Table 1 with our terms to include the model parameters. Second, we define model parameters as they come up in the results and then again in the discussion.*

The Discussion has been much improved with its more expansive engagement with the literature.

*Great to hear!*

Specific comments (line numbers refer to the track changes version):

L165: "(aboveground biomass R2=0.99, belowground NPP R2 165 =0.94)." I think you should include some rmse and/or bias information here too.

*Done*

L171: "but it also requires higher temporal resolution of data to be successful." This is a little simplistic. The information content / requirements for DA to be successful depends on multiple factors including, combination of data types, their uncertainty, the specific model in addition to data quantity. I think a more generalist statement on data richness would be more accurate.

*We now incorporate a more inclusive statement about data needs for data-model fusion to be successful.*

Figure 2 - this section of the analysis assumes a linear association between MAP, MAT and residence time. I suggest that some non-linear, threshold or bounding behaviour are also possible not best illustrated with this plot. A scatter plot would be more informative.

*We now include the raw relationships in Figure 2. Additionally, we tested goodness of fit between log-linear and linear models for each relationship and the linear models always produced the best fit.*

Figure 4 - I think this figure should go to the SI and be replaced with one which focuses more closely on the sensitive parameters. Doing so should improve the readability of the figure.

*Done. The old figure 4 is now Figure A10, and we focus on four parameters in the new Figure 4 – c5, c6, mscut, and Q10. We think this greatly improved the readability of this figure as the reviewer suggested.*